# Possible Causes of False General Relativity Violations in Gravitational Wave Observations

Anuradha Gupta[1]* K. G. Arun[2] Enrico Barausse[3,4] Laura Bernard[5] Emanuele Berti[6] Sajad A. Bhat[7,2] Alessandra Buonanno[8,9] Vitor Cardoso[10,11] Shun Yin Cheung[12,13] Teagan A. Clarke[12,13] Sayantani Datta[14,2] Arnab Dhani[8,15] Jose María Ezquiaga[10] Ish Gupta[15] Nir Guttman[12,13] Tanja Hinderer[16] Qian Hu[17] Justin Janquart[18,19] Nathan K. Johnson-McDaniel[1] Rahul Kashyap[15,20] N. V. Krishnendu[21] Paul D. Lasky[12,13] Andrew Lundgren[22] Elisa Maggio[8] Parthapratim Mahapatra[2] Andrea Maselli[23,24] Purnima Narayan[1] Alex B. Nielsen[25] Laura K. Nuttall[22] Paolo Pani[26] Lachlan Passenger[12,13] Ethan Payne[27,28] Lorenzo Pompili[8] Luca Reali[6] Pankaj Saini[2] Anuradha Samajdar[18,19] Shubhanshu Tiwari[29] Hui Tong[12,13] Chris Van Den Broeck[18,19] Kent Yagi[14] Huan Yang[30, 31, 32] Nicolás Yunes[33] B. S. Sathyaprakash[15,20,34]

**1** Department of Physics and Astronomy, The University of Mississippi, University, Mississippi 38677, USA

**2** Chennai Mathematical Institute, Siruseri 603103, India

**3** SISSA, Via Bonomea 265, 34136 Trieste, Italy and INFN Sezione di Trieste

**4** IFPU - Institute for Fundamental Physics of the Universe, Via Beirut 2, 34014 Trieste, Italy

**5** Laboratoire Univers et Théories, Observatoire de Paris, Université PSL, Université Paris Cité, CNRS, F-92190 Meudon, France

**6** William H. Miller III Department of Physics and Astronomy, Johns Hopkins University, Baltimore, Maryland 21218, USA

**7** Inter-University Centre for Astronomy and Astrophysics, Post Bag 4, Ganeshkhind, Pune 411007, India

**8** Max Planck Institute for Gravitational Physics (Albert Einstein Institute), Am Mühlenberg 1, Potsdam 14476, Germany

**9** Department of Physics, University of Maryland, College Park, MD 20742, USA

**10** Niels Bohr International Academy, Niels Bohr Institute, Blegdamsvej 17, 2100 Copenhagen, Denmark

**11** CENTRA, Departamento de Física, Instituto Superior Técnico – IST, Universidade de Lisboa – UL, Avenida Rovisco Pais 1, 1049-001 Lisboa, Portugal

**12** School of Physics and Astronomy, Monash University, VIC 3800, Australia

**13** OzGrav: The ARC Centre of Excellence for Gravitational-Wave Discovery, Clayton, VIC 3800, Australia

**14** Department of Physics, University of Virginia, Charlottesville, Virginia 22904, USA

**15** Institute for Gravitation and the Cosmos and Physics Department, Penn State University, University Park PA 16802, USA

**16** Institute for Theoretical Physics, Utrecht University, Princetonplein 5, 3584 CC Utrecht, The Netherlands, EU

**17** Institute for Gravitational Research, School of Physics and Astronomy, University of Glasgow, Glasgow, G12 8QQ, United Kingdom

**18** Institute for Gravitational and Subatomic Physics (GRASP), Utrecht University, Princetonplein 1, 3584 CC Utrecht, Netherlands

**19** Nikhef – National Institute for Subatomic Physics, Science Park, 1098 XG Amsterdam, The Netherlands

**20** Department of Astronomy and Astrophysics, Penn State University, University Park PA 16802, USA

**21** International Centre for Theoretical Sciences, Survey No. 151, Shivakote, Hesaraghatta, Uttarahalli, Bengaluru, 560089, India

**22** University of Portsmouth, Portsmouth, PO1 3FX, United Kingdom

**23** Gran Sasso Science Institute (GSSI), I-67100 L'Aquila, Italy
**24** INFN, Laboratori Nazionali del Gran Sasso, I-67100 Assergi, Italy
**25** Department of Mathematics and Physics, University of Stavanger, NO-4036, Norway
**26** Dipartimento di Fisica, "Sapienza" Universitá di Roma & Sezione INFN Roma1, Piazzale Aldo Moro 5, 00185, Roma, Italy
**27** Department of Physics, California Institute of Technology, Pasadena, CA 91125, USA
**28** LIGO Laboratory, California Institute of Technology, Pasadena, CA 91125, USA
**29** Physik-Institut, Universität Zürich, Winterthurerstrasse 190, 8057 Zürich, Switzerland
**30** Department of Astronomy, Tsinghua University, Beijing 100084, China
**31** Perimeter Institute for Theoretical Physics, Waterloo, ON N2L2Y5, Canada
**32** University of Guelph, Guelph, Ontario N1G 2W1, Canada
**33** Illinois Center for Advanced Studies of the Universe & Department of Physics, University of Illinois at Urbana-Champaign, Urbana, Illinois 61801, USA
**34** School of Physics and Astronomy, Cardiff University, Cardiff, CF24 3AA, United Kingdom

⋆ agupta1@olemiss.edu

## Abstract

**General relativity (GR) has proven to be a highly successful theory of gravity since its inception. The theory has thrivingly passed numerous experimental tests, predominantly in weak gravity, low relative speeds, and linear regimes, but also in the strong-field and very low-speed regimes with binary pulsars. Observable gravitational waves (GWs) originate from regions of spacetime where gravity is extremely strong, making them a unique tool for testing GR, in previously inaccessible regions of large curvature, relativistic speeds, and strong gravity. Since their first detection, GWs have been extensively used to test GR, but no deviations have been found so far. Given GR's tremendous success in explaining current astronomical observations and laboratory experiments, accepting any deviation from it requires a very high level of statistical confidence and consistency of the deviation across GW sources. In this paper, we compile a comprehensive list of potential causes that can lead to a false identification of a GR violation in standard tests of GR on data from current and future ground-based GW detectors. These causes include detector noise, signal overlaps, gaps in the data, detector calibration, source model inaccuracy, missing physics in the source and in the underlying environment model, source misidentification, and mismodeling of the astrophysical population. We also provide a rough estimate of when each of these causes will become important for tests of GR for different detector sensitivities. We argue that each of these causes should be thoroughly investigated, quantified, and ruled out before claiming a GR violation in GW observations.**

Received Date
Accepted Date
Published Date

## Contents

# 1   Introduction

Einstein's general theory of relativity (GR) stands as the most successful theory of gravity to date. Rigorously tested in weak-field, low-speed, and linear gravity regimes, GR has consistently withstood all scrutiny. Gravitational waves (GWs) are predictions of GR and offer a unique avenue for exploring spacetime dynamics in extreme gravitational conditions. Despite the widespread use of GWs from compact binary coalescences (CBCs) for testing GR, no deviations from the theory have been found so far (e.g., [1–12]).

The sensitivity of GW detectors has been continuously improving and LIGO and Virgo detectors are currently witnessing their fourth observing run (O4) with Advanced LIGO and Virgo sensitivity [13] which later will be joined by KAGRA [14]. These detectors will be further upgraded for the fifth observing run (O5) during 2027-2029 [15] with A+ sensitivity [16], and they will eventually be joined by LIGO-India [17,18]. Looking further into the future beyond

O5, there is a possibility for detectors with $A^{\#}$ sensitivity [19] that are expected to be twice as sensitive as A+. Moreover, there are concrete plans to build next generation (XG) detectors, such as Cosmic Explorer [20] and Einstein Telescope [21], that are expected to be at least 10 times more sensitive than the current detectors in O4. The first space-borne mission, LISA [22], is scheduled to be launched in the mid-2030s, and it might be followed by other missions such as TianQin [23, 24], Taiji [25], DECIGO [26, 27] and LGWA [28].

With these improvements in sensitivity, thousands of CBCs are expected to be observed with high signal-to-noise ratios (SNRs) [16]. A subset of these mergers will cover extreme regions of the parameter space, including highly spinning and/or strongly precessing binaries, binaries with eccentricity, binaries involving dense matter, etc. Such binaries will have the capability to test GR stringently and constrain beyond-GR effects, if present in the data. For example, higher black hole spins lead to higher curvature outside the horizon [29], which allows one to place constraints on a variety of higher-derivative or curvature-corrected theories [30, 31]. More so, the near-horizon region of black holes could potentially access energies as large as the Planck scale that could alter the black hole ringdown spectrum if GR is modified near the event horizon [32, 33]. There is also the possibility that GR may be violated not in the ultraviolet (UV), but rather in the infrared (IR) regime of the theory, aimed at offering an alternative explanation of the dark sector. In this "IR" scenario, extending the reach of GW detectors to lower frequencies may help observe possible deviations from GR in the inspiral phase of CBCs [34–37].

The majority of tests of GR currently performed rely on waveform models that are compared with the GW data. Often these tests are formulated as *null tests* where one looks for possible departures from GR by introducing deviation parameters on a given waveform model. No statistically significant deviation from GR has been observed at the level of individual events or for the whole population [5]. However, there were a couple of events in GWTC-3 [38] that suggested GR deviations, though further investigations are needed since these deviations could be due to the use of imperfect waveform models or inadequately understood noise artifacts in the data [39].

Due to the complexity of the physics of compact binary mergers as well as the detector noise modeling, it is extremely important that there is a consensus in the community about the necessary conditions that will warrant a much more comprehensive list of tests to be carried out to vet (or rule out) a potential GR violation claim. There are two aspects to this issue. The first is to identify all possible causes which might lead to a false GR violation. The second is a checklist to be executed upon encountering a strong candidate for GR violation. The objective of this paper is to tackle the first aspect and enumerate an extensive list of scenarios that may appear as violations of GR, when in fact they are not. The second aspect requires us to construct a checklist of items that address other issues such as the statistical significance of the violation, the status of the detector, or if the violation is in contradiction with other experiments or astrophysical observations. A companion paper will address these issues and a possible formulation of a GR violation detection checklist. It is worth noting that a similar effort has been made in Section 7 of [40], albeit in the context of tests of GR using LISA. Our goal here is to broadly classify different effects that can mimic a GR violation in the context of present- and next-generation ground-based interferometric observational facilities.

There are at least three distinct scenarios that can mimic a GR violation (see Fig. 1): noise artifacts in data, waveform systematics, and astrophysical aspects, each of which is discussed at length below. Much work has already been done to understand aspects of these scenarios on tests of GR. Broadly speaking, these three scenarios also have the possibility to impact other scientific conclusions based on GW data, such as constraints on astrophysical sources or cosmological models. In many cases, efforts to understand the impact of these scenarios on astrophysics or cosmology can also illuminate potential impacts on tests of GR.

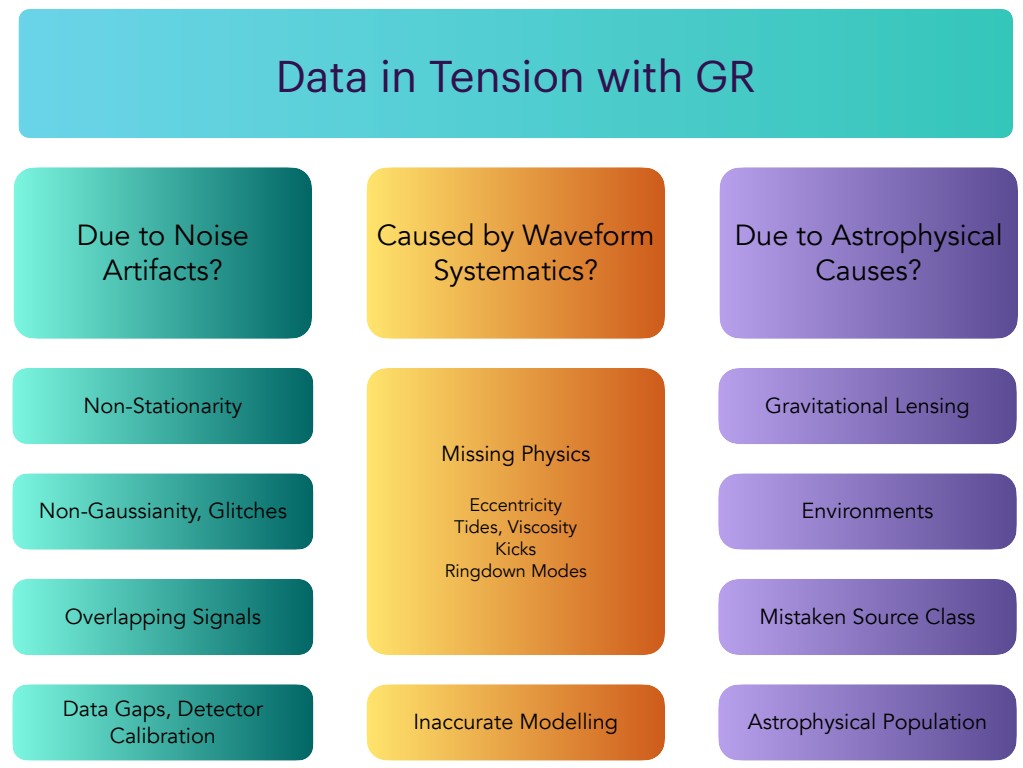

Figure 1: The diagram illustrates the principal false causes of GR violation in GW data. They are classified into three main classes: (a) noise artifacts, (b) waveform systematics, and (c) astrophysical effects.

To keep the discussion coherent, we group the causes only into these three scenarios even if this classification, or the distinction between any two causes, may seem somewhat arbitrary. For example, we keep the *overlapping signals* under noise artifacts even if this is not, strictly speaking, an instrumental noise source. Similarly, we divide issues related to waveform systematics into two main themes (*missing physics* and *inaccurate modeling*), even if the distinction between the two is not always obvious. By "missing physics" we mean cases when a particular effect is not included at all, or only partially included in the waveform models (e.g., tides and higher-order ringdown modes), while "inaccurate modeling" refers to intrinsic limitations of the waveform models in fully describing the known features of GR (e.g., waveform truncation errors).

While most of the scenarios discussed below could lead to confusion with a GR violation in a given event or subset of events, any GR deviation should be consistent across the dataset, e.g., a given theory should explain why there is evidence for deviations in certain events and not in others in a similar region of the parameter space. The ever-increasing number of events expected in the future will help sort out these situations.

## 2   Noise Systematics

Current interferometric GW detectors are limited by fundamental noise sources [13] which causes the noise to appear as stationary and Gaussian only over short time scales and ranges of frequency [41]. In reality, however, noise from the detectors is neither Gaussian nor stationary (see, e.g., [41–43]). It can be relatively easy to spot times of extremely bad data quality in GW data, but the challenge lies with times of subtle data quality issues. The origin of noise sources

117 is notoriously difficult to pinpoint, even for obvious cases of poor data quality. However, it is
118 essential that we understand our noise, remove any bias that noise introduces, and accurately
119 infer the parameters of the observed sources.

120     In this Section, we discuss the three main sources of noise (namely, non-stationary, non-
121 Gaussian, and overlapping signals) observed in ground-based detectors that can affect our
122 inference of transient GW signals. We also discuss the systematic error due to the gaps in
123 data and calibration of the GW instruments that may also introduce some bias in the inference
124 results.

## 2.1  Non-stationarity

126 Non-stationarity is a broadband form of noise which causes the statistical properties of the
127 background to change with time. Non-stationarity occurs on the order of tens of seconds
128 in the current LIGO detectors and can be caused by both instrumental and environmental
129 sources [42,44], such as detector's lockloss [45], variable seismic motion, thunderstorms [43],
130 magnetic effects from lightning strokes [46], and radio frequency interference [42, 43]. This
131 form of noise has been shown to affect the estimation of source parameters [47,48]. Modelled
132 searches typically estimate a detector's power spectrum over several minutes [49–51], which
133 can cause the matched filter to miss the variable nature of the noise, affecting the search sensi-
134 tivity. One method to account for this is to construct a statistic which tracks the variation of the
135 power spectrum and to normalize the ranking statistic used by the detection pipeline [51–53].
136 The method presented in [52] is also used to assess the stationarity of the data around can-
137 didate GW events [43]. This is because non-stationary noise can impact binary neutron star
138 signal parameters [54,55] since noise estimates, usually calculated over minutes, fail to cap-
139 ture variations on shorter time scales. As signals from sufficiently massive binary black holes
140 are usually shorter than the typical time scale of non-stationary noise, these sources are not
141 thought to be affected. However, due to their long duration in the sensitivity band, sub-solar
142 mass binary black hole searches will be affected, especially in the XG era where the signal
143 duration could be up to several days.

144     To date, this form of noise has not seriously affected the conclusions drawn from any of the
145 LIGO-Virgo-KAGRA collaboration's GW events. However, it could be an issue in the future, and
146 certainly for XG detectors which will be more sensitive to noise variability and observe hours-
147 long signals, breaking the assumption of stationarity. As such, future methods for detecting
148 and interpreting GW signals should account for the variable nature of the detector noise.

## 2.2  Noise Transients or Glitches

150 Transient noise artifacts, also known as glitches, are also a common problem in interferometric
151 GW detectors. Glitches can mask or mimic a signal and add to the noise background of tran-
152 sient GW searches (see, e.g., [42,43,56]). Glitches occur frequently in all detectors; in the third
153 observing run, the rate of glitches was between 0.29 to 0.32 per minute for LIGO-Hanford,
154 1.10 to 1.17 per minute for LIGO-Livingston and 0.47 to 1.11 per minute for Virgo [38]. The
155 inferred population properties of glitches have been shown to typically exhibit characteristics
156 similar to CBC signals with large mass ratios and large spins, in contrast to the observed as-
157 trophysical properties, which tend to have near equal masses and moderate spins [57]. This is
158 because CBC signals with large mass ratios and large spins can have more 'irregular' waveform
159 morphologies compared to equal-mass, non-spinning CBCs from the twisting-up effects due to
160 precession. Therefore, this class of signals has a better chance of fitting well with the terrestrial
161 disturbance produced by glitches that lack a CBC signal's typical chirping-up characteristics.

162     The morphology of glitches, in particular their time duration and the frequency space
163 they affect, can be highly variable between different glitch classes. For example, blip glitches

164 (e.g., [58]) are fractions of a second in duration, covering a large bandwidth (e.g., tens to
165 hundreds of Hz) and can mimic a GW signal of high mass compact binaries. We still do not
166 know the origin of these types of glitches as they do not have a known environmental or
167 instrumental coupling, but they appear to have different subcategories that may be caused by
168 different physical mechanisms. In the third observing run, these types of glitches occurred
169 4 times per hour at LIGO-Livingston and twice per hour at LIGO Hanford [43]. However,
170 scattering glitches (e.g., [59]) caused by microseism noise, can be a few seconds long, and
171 present as arches in the time-frequency plane, affecting frequencies below 100 Hz. These
172 glitches manifest due to a small fraction of laser light scattering off a test mass, hitting a moving
173 surface, and recombining with the main beam. These types of glitches are most prevalent when
174 the ground motion is high. As such they can seriously contaminate hours of data, but not be
175 a concern for weeks at a time.

176 Tracking the occurrence and emergence of new glitch types can be a challenge. Both LIGO
177 and Virgo take advantage of machine learning frameworks, combined with citizen scientists,
178 to classify glitches based on their morphology in the time-frequency plane. GravitySpy [60]
179 has been in operation since the second observing run, and citizens have helped to classify
180 LIGO glitches into 23 distinct classes [43]. GWitchHunters [61] helps to classify glitches from
181 the Virgo detector, and has been open to the public since November 2021. Both projects will
182 prove extremely valuable in identifying and understanding glitches in the fourth and future
183 observing runs.

184 Glitches overlapping or being in the vicinity of a real GW signal can be a huge problem.
185 In fact, in the third observing run 24% of GW events had a glitch within the analysis window
186 for one or more detectors [38]. These glitches did not impact the detection of these events,
187 but they had to be mitigated before the source parameters could be accurately estimated. A
188 prime example of this issue first arose in the interpretation of GW170817 where a short glitch
189 occurred 1.1 seconds before the coalescence of the event, lasting only 5 ms [62]. Nonetheless,
190 this noise had to be removed before the parameters of the event could be accurately deter-
191 mined. Macas *et al.* [63], for example, shows that certain types of glitches can cause the sky
192 localization to be incorrectly determined for certain types of signals, which can even affect
193 follow-up with large field of view telescopes (i.e., 20 deg$^2$).

194 There are a number of ways in which noise can be removed or subtracted from the data.
195 Should the noise be broadband in origin then noise subtraction over the course of hours or
196 days is needed. This can be achieved using auxiliary channels which monitor noise sources
197 at different points around an interferometer. A coupling function can then be determined
198 to understand how much a certain type of noise affects the GW channel, and the noise sub-
199 tracted [64,65]. This method is optimal when the data are Gaussian and stationary. More re-
200 cent work has focused on machine learning techniques to cope with data with non-stationary
201 noise couplings [66].

202 For short instances of transient noise that may be in the vicinity of an event, there are
203 a few methods which are currently used. A window function can be applied to zero out the
204 glitch; this method is known as gating [49,67]. Gating has the benefit of being quick, however
205 uncontaminated data will also be removed using this method, as the window function needs
206 to be smoothly applied to avoid adding filtering artifacts to the data. Hence, this method is
207 not appropriate if the glitch is not well localized in time and is close to an event's coalescence
208 time. A more robust method is to model a glitch with a time-frequency wavelet reconstruc-
209 tion and use this to subtract it from the data; this method is applied using the BayesWave
210 algorithm [68]. This method has been used to great effect in the third observing run [38]. An-
211 other method, called gwsubtract, uses data from an auxiliary witness to the noise to subtract
212 the noise from the GW channel [64,69]. This was done for the first time around the event
213 GW200129 [38], which seems to exhibit characteristics consistent with spin induced orbital

precession [70]. However, Payne *et al.* [71] find that residual data quality issues leftover from this cleaning process may be the origin of the precession observed in GW200129. Moreover, in a ringdown analysis of GW200129 [39] found a deviation from GR in the peak of the GW amplitude while employing a nonprecessing `SEOBNRv4HM_PA` model [72–74] but they ascribe it to waveform systematics (modeling of spin precession) or data-quality issues (glitch mitigation procedures). Regardless, this example of GW200129 highlights the complexities and care that need to be taken when removing glitches from GW data and interpreting results from inference analyses.

Glitches will always remain a feature of GW data because as the detector sensitivity improves noise artifacts that were sub-dominant will become more relevant. It is unfeasible to remove them all. New methods are being developed to effectively deduce both source and population parameters by integrating realistic but imperfect data. For example, Ashton *et al.* [75] uses Gaussian processes to replace the traditional GW likelihood. This method, in principle, can model arbitrarily colored noise, non-stationarity, and glitches, to augment the approach to estimate the parameters of sources. In addition, Heinzel *et al.* [76] presents a method for inferring the population of GW sources contaminated by blip glitches. They are able to infer the shape parameters of a GW population, whilst simultaneously inferring the population of the glitch background events.

In order to be confident that a signal is indeed a violation of GR, characteristics that may arise due to the noise identified here need to be understood. Work has started in this regard, for example with [77]. They investigated how an overlapping binary black hole signal with three different glitches can affect tests of GR before and after the glitches were mitigated. Moreover, they only considered a glitch in a single detector out of three and still found a GR deviation when the glitch was not mitigated. The authors also point out that their study is not sufficient to give quantitative statements about the effects of certain glitch classes or mitigation methods on tests of GR. Therefore, their work needs to be extended to assess the amount of GR deviation in different realizations of Gaussian noise, the effect of non-stationarities in the noise background, and the effect of data cleaning methods on mimicking GR deviations.

## 2.3 Contamination from Overlapping Signals

As the sensitivity of ground-based GW detectors improves, the chances of observing time-overlapping signals will also increase [78]. This may demand a shift in our detection and parameter estimation strategies since current pipelines, designed for single GW signals, may yield biased results when applied to overlapping signals. However, several studies have shown that the detection [79, 80] and parameter estimation [78, 81–83] of overlapping signals are not a significant concern. For example, [79] and [80] showed that it is possible to detect and discern overlapping signals from binary neutron stars using a matched filtering algorithm. Relton *et al.* [84] conducted a more thorough study with both modeled and unmodelled search analyses and found that both analyses can detect overlapping signals from binary black holes when merging $> 1$ s apart. However, unmodelled analysis can identify overlapping signals merging within $< 1$ s while modeled analysis can only identify only one of the two overlapping signals. Himemoto *et al.* [83] thoroughly explored the parameter space and concluded that overlapping signals do not lead to large biases in parameter estimation provided the coalescence times and redshifted chirp masses of the two overlapping signals differ by at least $10^{-2}$ s and $10^{-4} M_{\odot}$ for binary neutron star mergers and $10^{-1}$ s and $10^{-1} M_{\odot}$ for binary black hole mergers, respectively.

Nonetheless, overlapping signals do pose biases in both detection and parameter estimation of sources and methods have been proposed to correct those biases [85–88]. For example, Wu & Nitz [85] pointed out that overlapping signals reduce the search sensitivity by changing the noise's amplitude spectral density and proposed an updated search campaign on overlap-

ping signals using the single-detector signal subtraction method. Johnson *et al.* [89] pointed out that the presence of overlapping signals may require us to revisit the definition of the likelihood as well as the assumption that source confusion can be treated as stationary Gaussian noise. Possible remedies to the bias in source parameter inference have been suggested, either from a Fisher Matrix study [86] or adapting the signal model accordingly in the Bayesian likelihood [87]. Langendorff *et al.* [88] used normalizing flows as an avenue to deal with the computational burden coming from multiple-signal analyses in case of overlaps.

Moreover, Hu & Veitch [90] studied the effects of waveform inaccuracy and overlapping signals on tests of GR and demonstrated that when combining results from multiple signals with overlaps, the deviation from GR decreases when waveform models are perfect (no waveform inaccuracy), but inaccurate waveform modeling can lead to a false deviation of GR for overlapping signals. Dang *et al.* [91] extended this study to higher post-Newtonian (PN) deformation parameters. They concluded that although a non-negligible number of overlapping signals can lead to false GR violations at the individual event level, when the results are combined, the biases tend to smoothen out, leading to a preference for GR at the population level inference. (We discuss the effects of population-level analyses on tests of GR in more detail in Section 4.4.)

All these studies focussed on overlaps arising in the data of XG detectors, since the probability of observing overlapping signals in the era of A+ sensitivity [16] or Voyager [92] is very small [78]. However, it is likely that a quiet GW signal below the detection threshold is present along with the dominant GW signal in the data [93]. This will not pose a problem for estimating individual source parameters, but issues may arise when combining multiple signals, where sub-threshold events collectively act as background or confusion noise [94,95]. Although [94,95] considered signals in the XG era only, we might need to consider the effect of a confusion-noise-like background in O5 or $A^{\#}$ era in the context of testing GR. Moreover, quieter signals may result in imperfect subtraction of the GW model from data when following the definition of likelihood to infer source properties under the assumption of stationary, Gaussian noise. Consequently, combining results across multiple signals to infer population properties could gradually accumulate biases from each single-signal analysis, potentially mimicking noise properties [89] and introducing deviations from GR.

## 2.4 Gaps in the Data

The data we expect to collect from XG detectors is likely to contain gaps, due to loss of lock at the inferometers that could be caused by a plethora of instrumental or anthropomorphic reasons. The sensitivity band of current detectors is such that GW signals are in the band for about 30 minutes at most. The likelihood of a data gap in such a short window is small, and if it occurs, it is likely to decrease the SNR significantly, since the recovery time (for the instrument to reacquire lock and start data taking again) is comparable to the signal duration. This scenario changes drastically with XG detectors because the low-frequency sensitivity is greatly increased, allowing for the observation of signals for many hours to days. The likelihood of a data gap in this window is larger, and if it occurs, it is likely to both decrease the SNR of the event and deteriorate the analysis of the GW source.

Not much work has gone yet to study the effect of data gaps in XG detectors, but some work already exists for data gaps in space-based detectors, from which we can extrapolate some conclusions. Previous work has shown that data gaps can deteriorate and bias parameter estimation for certain sources [96, 97], in particular when the data gap coincides with the merger phase. In general, we would expect that a data gap during the merger would inhibit our ability to constrain deviations from GR at high PN order, while gaps in the early inspiral will be the same for low (or negative) PN order modifications to GR. In particular, if the data has a gap, but our analysis does not account for it, parameter correlations between

non-GR and GR parameters are likely to introduce biases that may lead to a false GR violation. Certain methods, such as Bayesian data augmentation [98], however, can be used to include missing data periods as auxiliary variables when sampling the posterior distribution of model parameters that have shown promise at eliminating biases.

## 2.5 Detector Calibration Error

The GW strain data $d$ are not directly recorded by the interferometer. Instead, it is reconstructed from the voltage $v(f)$ measured by photodetectors and a response function $R(f)$ that relates the digital readout and GW strain, i.e., $d(f) = R(f)v(f)$ [99]. The calibration process includes a series of measurements to construct a reference model for the response function [99–101]. Bias in any step of this process can lead to errors in the measured strain data, and systematic errors in parameter estimation could arise if the calibration error is not taken into account. Vitale *et al.* [102] investigate the consequences of calibration error in Bayesian inference of source parameters. By comparing the ratio between the calibration systematic errors and the statistical uncertainties, their results show that the parameters that suffer the largest biases are those mostly related to the amplitude of GW signals: on average, calibration systematic errors are of $\mathcal{O}(1/100)$ of the statistical uncertainty for sky location parameters and $\mathcal{O}(1/1000)$ for masses. This is potentially because phase errors are localized in frequency and do not accumulate over the inspiral. It implies that calibration errors could have a minor effect in parameterized tests of GR that modify the phase of waveform. Vitale *et al.* [102] also conclude that $< 20\%$ of amplitude calibration error or $< 10 - 20°$ of phase calibration error should not lead to significant biases for all but the strongest signals in the advanced LIGO era, consistent with [103] and [104]. Furthermore, they report that the calibration systematic error is not strongly correlated with SNR as the calibration affects both noise and signal. However, whether such a level of calibration systematics is tolerable in the XG era where SNR can reach $\mathcal{O}(1000)$ is worth investigating. It is still of great importance to improve calibration techniques along with the high sensitivity in the XG era [44, 105].

It is possible to quantify and mitigate calibration errors in detection and data analysis. The uncertainty of the response function can be indicated by the photon calibrators which apply a known radiation pressure directly on the test masses within the detector [99, 106–108]. Abbott *et al.* [109] reported $< 10\%$ calibration uncertainty in the strain amplitude and $< 5°$ in phase during the first LIGO-Virgo observing run , and in the third observing run these uncertainties were reduced to $< 7\%$ and $< 4°$, respectively [110]. Note that these are overall uncertainties and systematic errors alone are even smaller. These estimates of calibration uncertainties are used as priors to marginalize uncertainties in GW strains during parameter estimation, which effectively mitigates the calibration error [111, 112]. However, this technique might conceal tiny deviations from GR, since it marginalizes over some level of uncertainties on amplitude and phase. Hence, the effect of calibration errors on tests of GR needs to be studied for current and future GW detectors, so that it can be ruled out (or included) as one of the possible causes for false GR violations.

# 3 Waveform Systematics

## 3.1 Missing Physics in Waveform Models

The current state-of-the-art waveform models used in tests of GR still lack certain physical effects, such as eccentricity of the binary's orbit, overtones, and non-linearities in the ringdown phase of the binary merger, etc. Including each of these known physical effects individually is crucial for precision GR tests, but their collective inclusion is essential for unbiased assessments

357   of GR. Here we discuss missing physical effects that could lead to a false GR violation.

### 3.1.1 Eccentricity

The eccentricity of a binary's orbit depends on the formation history of the binary. Binaries formed through isolated formation channels in the galactic field are expected to have negligible eccentricity when observed in the frequency band of ground-based detectors, whereas binaries inside dense stellar environments such as globular clusters and nuclear star clusters might have moderate to high eccentricities when observed by these detectors. In an isolated formation channel [113], the binary goes through various mass transfer episodes between its components, and as the components evolve and undergo supernova explosions, the binary orbit could gain some eccentricity due to supernova kicks. However, due to the emission of gravitational radiation [114, 115] the binary's orbit shrinks, and the binary sheds away all its eccentricity over the long inspiral, leaving it with negligible eccentricity close to merger [114]. For example, if a binary with an initial orbital eccentricity of 0.2 emits GWs whose dominant mode has a frequency of 0.1 Hz, the eccentricity reduces to $\sim 10^{-3}$ when it reaches a dominant mode GW frequency of 10 Hz. That is why binaries detected by LIGO/Virgo are expected to be quasi-circular. On the other hand, a fraction of dynamically formed binaries can still have some eccentricity (and as high as $\sim 1$ at 10 Hz) when observed in the frequency band of the LIGO/Virgo detectors [116–124]. Further, environmental effects such as accretion and dynamical friction can also increase the eccentricity of binaries [125].

The problem of misinterpreting eccentricity as a potential GR violation is currently a two-fold problem. First, of missing physics; namely, the inclusion of both eccentricity, argument of periapsis (although see [126]), and precession in an inspiral-merger-ringdown waveform model. Distinguishing eccentricity from precession without waveforms that include both [127] introduces systematic biases in the estimated binary parameters [128–132] that could be misconstrued as false violations of GR [133–137]. Second, the current analysis methods are producing inconsistent results [126, 128, 129, 138–140] for the same events such as GW190521 [141].

Once the above two problems are solved, the problem of eccentricity reverts back to being one of waveform systematics discussed in more detail in Section 3.2.2 below. We anticipate larger waveform systematics in systems with higher eccentricities. However, these are not the ones for which eccentricity will manifest as a violation of GR, due to the large-amplitude modulations that are inconsistent with a quasi-circular inspiral.

### 3.1.2 Tidal Effects

Neutron stars and their mergers are characterized not only by strong gravity but also by extreme matter conditions. To explore how matter affects the space-time deformations around these stars, we need to understand the relation between the dynamical properties of matter and the behavior of strong gravity. Analytic methods are used to model the early inspiral phase of a neutron star binary merger, where neutron stars are approximated as massive point particles with small corrections due to finite-size effects [142–144]. However, close to the merger finite size effects become significant and numerical relativity (NR) simulations are required to capture them accurately [145–148]. Effective one body models achieve a nonperturbative resummation of the PN information on tidal effects into a complete framework [145, 149–154]; some reduced-order-model versions incorporate NR-calibrated tidal models [148, 155, 156] as also used in Phenomenological models.

The tidal deformation of bodies is directly proportional to the Riemann tensor and its derivatives, produced primarily by the energy-momentum distribution of the companion [157], which becomes the second derivatives of the Newtonian potential for the electric-type quadrupole effect in the Newtonian limit. However, such effects are observable in the GWs only if they

produce significant mass and current type multipole deformations of the neutron stars in a binary system. The dominant deformations come from the electric-type, $l = 2$ tidal deformation, which imprints primarily in the GW phase evolution. However, it is important to note that these tidal effects are relatively small and become more pronounced as the binary approaches merger. While these effects are subtle, their detection has already provided invaluable insights [62], and with the advent of more advanced detectors (such as XG), we can look forward to even more precise measurements in the near and far future [158–161].

The effects of the tidal field on neutron star matter are studied using observed GWs [2], however, such results are susceptible to waveform systematics and incomplete modeling of neutron star physics. For example, Refs. [162–164] show that the inference of tidal parameters with XG detectors can be significantly affected due to waveform systematics. Not including subdominant tidal effects, such as dynamical tides, which become important in the inspiral regime, can also lead to substantial biases in the estimation of tidal parameters [153, 154, 165, 166]. Likewise, the effects of spins on dynamical tides [167–170], other spin-tidal couplings [148, 171], spin-induced multipole effects [172–175], nonlinear tides [176], higher-order relativistic corrections, and the GW features of tidal disruption in cases with precessing spins [177] are examples of areas requiring further investigations. Further, XG detectors will be sensitive to the octupolar electric and quadrupolar magnetic tidal deformabilities, and not including them in the waveform might bias the measurements [178].

Resonant mode excitations may contribute distinct features in the waveform from the tidal effect considered in [157]. As the inspiraling orbit passes through the frequency of a certain characteristic mode, the resonant excitation of the mode must be compensated by the loss of the same amount of orbital energy, speeding up the following orbital evolution. The excitation of gravity modes [179–181], the interface mode [182–184] and gravitomagnetic mode [185–188] have been studied, where for the latter two cases the phase modulation may reach the level of $\mathcal{O}(10^{-2}) - \mathcal{O}(10^{-1})$ radians in the frequency band of ground-based detectors.

Inaccurate or missing physics in analytical and NR modeling due to thermodynamical transformation of nuclear matter during inspiral and post-merger leads to waveform systematics. Such effects include, but not limited to, viscosity [189–192], thermal effects [193–197], phase transition to hyperon condensates or quark matter and other such transformations (see, e.g., [198–203] and also see Section 4.3.2 for discussion of proposed exotic matter that has not been observed but, may have compactness close to black holes). As shown in [204–206], the viscous effect introduces a new dissipative channel that modifies the GW phase at 4PN order and higher. If not included in the modeling, a signal containing such a 4PN effect could be misinterpreted as a GR deviation at that PN order and at neighboring PN orders.

Similar effects during the post-merger evolution are subject to systematic bias which requires emphasis on accurate post-merger waveform model development. Currently, only a few post-merger models exist and can detect such effects only in the XG detectors [207–211]. There are also sources of bias in parameter estimation that are exclusive to data analysis challenges arising from noise systematics. For a minority of events, multiple overlapping signals and confusion background created by CBC mergers could potentially lead to a bias in tidal deformability as described in Section 2.3.

Additionally, GR predicts relations between the spin-induced quadrupole moment and the (quadrupolar, electric) tidal deformability [8, 212–214] and between tidal deformabilities of different multipolar order and parity [215] or between different tidal parameters in gravitational waveforms for binary neutron star mergers [216, 217] which are only mildly sensitive to the neutron star equation of state. These relations have been used in GW data analyses to reduce the number of search parameters [218, 219] but small equation-of-state variation in these relations can induce systematic biases. One could, however, use constraints on nuclear physics from neutron star observations available at the time to keep updating and reducing

the amount of variation in the relations. For example, such variation has been reduced by 50% after GW170817 and current systematic errors on the tidal deformabilities are subdominant than statistical errors until the A$^{\#}$ era [220]. Another way to reduce systematic biases due to the variation in quasi-universal relations is discussed by [159]. It should be noted that the subpercent accuracy in the universal relations will become important in deducing the correct equation of state and hence in the tests of GR for the sensitivities corresponding to XG detectors. Since alternative theories predict different relations, an independent measurement of the quantities in the universal relations can therefore be used as null tests of GR, circumventing potential degeneracy with unknown nuclear physics [212–214, 221–223]. While the spin-induced quadrupole moment is expected to be small for neutron stars, the magnetic tidal deformability could be measured by XG detectors [178] and might need to be included in the waveform models.

Besides testing GR, these relations can be used to disentangle source misidentification (discussed in detail in Section 4.3.2), since each model of exotic compact objects other than neutron stars would display their own quasi-universal relation [222, 223]. Notably, the tidal deformability parameter may carry information about the nuclear equation of state and hence offer a unique tool to distinguish conventional neutron stars from the ones with exotic signatures. Analyzing binary neutron star mergers with exotic matter while using waveforms of conventional neutron star binaries could lead to false indications of GR violations. This needs to be investigated thoroughly, so that this effect could be ruled out or observed.

Assuming that our NR-assisted waveform models are accurate and free of systematic biases including those arising from the unknown equation of state, any deviation from the predictions will be indicative of either GR not being the complete theory of gravity or deviations in the coupling of matter to gravity, a subset of which is the test of the strong equivalence principle [224–229]. Therefore, only after ruling out the systematic effects arising from these inaccuracies, robust conclusions can be drawn about deviations from GR.

### 3.1.3 Kick-induced Effects

The anisotropic emission of GWs during a CBC carries away linear momentum and results in a recoil or *kick* of the merger remnant [230, 231]. The kick leaves the following imprints in the GW signal: the Doppler effect [232] and the aberration effect [233] on the post-merger signal along with an additional contribution of a (linear) memory effect [232] to the whole GW signal [234]. Since the black hole kicks are non-relativistic, the kick-induced effects are small and might not be important for current GW detectors but could be crucial for XG detectors [234, 235]. For loud ringdown signals (SNR$\gtrsim$ 100, [235]) in the XG era, these kick-induced effects, if not accounted appropriately in the waveform model [236, 237], might contaminate those tests of GR that depend on the post-merger signal and kick [238] of the remnant (see, e.g., [39, 234, 239–242]).

### 3.1.4 Beyond Fundamental Modes in Ringdown Signal

The gravitational radiation from a perturbed black hole is in the form of quasi-normal modes [243, 244]. At sufficiently late times following a binary black hole merger, it is expected that the remnant can be very well approximated by a perturbed Kerr black hole. Moreover, it is well known that the radiation at this stage is dominated by just the fundamental quasi-normal mode, since it is the slowest damped quasi-normal-mode (QNM) [245–247]. The frequency and damping time of a mode are in one-to-one correspondence with the remnant mass and spin. In principle, assuming GR and using NR simulations, the latter quantities could be predicted from the properties of the progenitor binary, which can be extracted from the premerger signal. In practice, waveform systematics in the premerger phase could jeopardize

this ringdown consistency test [248]. For example, large unmodelled eccentricity could lead to an inconsistency in the final mass and spin, and hence to a false GR deviation [135]. In the spirit of the original black-hole spectroscopy program [245–247, 249], it is therefore better to test GR using ringdown signals only, and an "agnostic" selection of multiple modes to model the ringdown [250].

Recently, there have been efforts to increase the range of validity of linear perturbation theory by modeling the early postmerger signal using overtones and mirror modes [250–262]. These studies show that the inclusion of these additional QNMs improve the remnant mass and spin estimates using a ringdown model. They also show that there will be biases in the remnant parameters if a ringdown model is used to describe early postmerger without the inclusion of such QNMs. Such biases in parameter estimation can show a deviation from the predictions of GR. Isi and & Farr [263] investigated the impact of an incomplete ringdown model on parameter recovery by analyzing a synthetic signal mimicking a binary black hole ringdown (see also [250] for a discussion). Their findings reveal biased parameter measurements in instances of very high ringdown SNR. Dhani & Sathyaprakash [255] displayed the modulations in the odd-$m$ modes in the waveform and how the inclusion of mirror modes in the ringdown waveform model can explain these modulations.

The BH spectroscopy program in GW literature aims to test the Kerr nature from the observed QNM spectrum. These tests are typically referred to as "no-hair" theorem tests too. However, since the tests are based on QNMs as the only observables, they are not sensitive to the type of BH hair — namely, primary hair[1] or secondary hair [264]. Therefore, any modification to the Kerr QNM spectra would fall under these tests. There are claims in the literature that overtones have been detected [265–267] and used to test the "no-hair" theorem with GW150914 [241]. However, there is a disagreement in the literature regarding the significance of the measurement of the first overtone in GW150914 [266, 268–271]. There are also theoretical arguments suggesting caution in the use of overtones for no-hair theorem tests [250, 259, 272–275]. The above authors show, using toy models, black hole perturbation theory and NR simulations, that even though the estimates of the final mass and spin of the black hole can be improved starting the ringdown analysis at earlier times by the addition of overtones, a linear model including only overtones is not appropriate at early times (see also [276]). Therefore, they contend that overtones are unphysical and that their role in a waveform model is to "fit away" other features in the signal, namely, transients related to the initial data, power-law tails at late times, and nonlinearities.

However, for less symmetric binaries than GW150914 (as commonly expected among current and future catalogs) the original black-hole spectroscopy program can be realized using higher-order modes in addition to the least damped QNM, i.e., $(l, |m|) = (3, 3), (2, 1), (4, 4)$, can be used to perform independent tests of the no-hair theorem [242, 270, 277–283]. Given current estimates of the merger rates, XG detectors are predicted to perform percent-accuracy tests for a few events per year [278, 283–285].

To conduct any of the above tests of GR using the perturbative ringdown model, one must make a choice on the start time of the ringdown to begin fitting exponentially damped sinusoids. The analysis should begin as soon as the perturbative prescription is relevant. On one hand, waiting too long to begin the analysis will make testing GR impossible because the strain amplitude has decayed exponentially (e.g., [286, 287]). However, beginning the analysis too early could result in overfitting to non-linear features in the signal (e.g., [250, 288]). To undertake robust tests of GR, some criterion for the analysis start time should be established through, e.g., searching for the earliest time at which one can measure self-consistent QNM

---

[1]In this context, primary hair refers to extra charges that are independent of the BH mass and spin (e.g., the electric charge in the Kerr-Newman solution), whereas secondary hair refers to extra charges that are fixed in terms of the mass and spin.

parameters with time [259, 260, 262]. A further source of systematics is the decomposition of QNMs in spherical rather than spheroidal harmonics; if unmodelled, the spherical-spheroidal mode mixing introduces biases for highly spinning remnants [250].

Another important effect of the nonlinearity in the ringdown stage is the presence of second-order QNMs [289–291], which are generated through mode-mode couplings. The frequency of a second-order QNM is twice as the associated "parent" linear QNM. Its amplitude and phase are also uniquely determined by the linear mode [292–294], as a nontrivial prediction of GR at the nonlinear level. The dominant nonlinear modes may be observable with XG detectors, although event rates are uncertain [295].

An approach complementary to null tests using QNM frequencies and damping times is to test QNM amplitude-phase relations predicted by NR simulations within GR. This test was successfully applied to GW190521 in [296], finding that measurement errors for this event are still large, but would strongly improve for the louder detections routinely expected for XG detectors.

Finally, because of its short duration, one should be careful with the statistical methods and their underlying assumptions while analyzing the ringdown signal. Seemingly innocuous data processing choices such as the uncertain starting time, duration of the signal, and noise estimation techniques can lead to materially different inferences [241, 268, 269, 297–299]. While the ringdown signal is typically analyzed in the time domain, frequency domain methods have also been proposed [257, 269, 300, 301] with the approach of [300] shown to be formally equivalent to the time-domain approach [263]. Even then, [300] comes to a different conclusion regarding the ringdown of GW190521 compared to [4] or [302]. This highlights the need to better understand systematics and data analysis techniques in the analysis of ringdown signals.

## 3.2 Inaccurate Modeling of Known Physics in Quasi-Circular Waveform Models

### 3.2.1 Higher-order Modes, Precession, and Memory

Gravitational waveforms can be decomposed in the basis of spin weighted spherical harmonics with spin weight $s = -2$, $Y_{-2}^{lm}(\iota)$, where $\iota$ is the inclination angle. In this basis, for nonprecessing systems, the dominant contribution to the GW amplitude comes from the $(l, |m|) = (2, 2)$ harmonics. The $(2, 1)$ and $(3, 3)$ harmonics are subdominant and suppressed by a prefactor that goes to 0 for symmetric (equal mass) binaries [303–307]. These modes only contribute for systems that are not face-on/off ($\iota \neq 0, 2\pi$), and become particularly important for unequal mass binaries. The presence of these higher-order modes causes characteristic modulations in the amplitude and phase of the waveform.

The effect of higher-order modes becomes even more important in the presence of spin-induced precession. Spin-induced precession occurs when the spin angular momentum vectors of the binary components are not aligned with the orbital angular momentum vector, leading to the precession of the orbital angular momentum (or, equivalently, the orbital plane of the binary) as well as the spin vectors about the total angular momentum of the binary. The effect of precession is best understood by considering two frames of reference [308–310]— the *inertial* frame in which the binary appears to be precessing, and the *co-precessing* frame that follows the instantaneous motion of the orbital plane where the effects of precession disappear. The inertial modes can then be approximately described as the sum of nonprecessing modes with the same $l$ value and all possible $m$ values, each rotated using Wigner D-matrices which depend on the instantaneous position of the orbital plane [311]. Thus, due to spin-induced precession, subdominant precessing modes will have contributions from both dominant and subdominant nonprecessing modes, increasing the precession effect due to the presence of higher-order modes in the waveform [312].

A consequence of using nonprecessing modes to approximate the co-precessing-frame signal is that these obey the reflection symmetry $h_{\ell m} = (-1)^{\ell} h_{\ell -m}^*$, which no longer holds for precessing binaries [313, 314]. Most state-of-the-art waveform models, with the exception of NRSur7dq4 [237] and IMRPhenomXO4a [315, 316], currently rely on this approximation. While the impact of anti-symmetric contributions to the waveform modes is typically small, neglecting these effects could result in biased measurements of the spin magnitude and orientation at high SNR [317, 318].

Currently, state-of-the-art nonprecessing waveforms like IMRPhenomXHM [319] include the harmonics $(l, |m|) = (2, 1), (3, 3), (3, 2), (4, 4)$, and SEOBNRv5HM [320], in addition to these, also includes $(l, |m|) = (4, 3)$ and $(5, 5)$. Their precessing counterparts are IMRPhenomXPHM [321] and SEOBNRv5PHM [322], respectively. The widely used NR surrogate waveform model, NRSur7dq4, has been trained with simulations with mass ratio less than 4, and contains all spherical-harmonic modes with $l \leq 4$.

Many studies have explored the improvement in the inference of source parameters due to the inclusion of spin-induced orbital precession and higher-order modes [323–326]. Particularly, for edge-on systems, including higher-order modes improves parameter estimation by breaking the luminosity distance-inclination angle degeneracy, whereas modulations due to spin-induced precession break the degeneracy between the spin and mass parameters. Additionally, the amplitude of the higher-order modes also brings information about the mass ratio of the source.

We should note that none of these models discussed above contain the memory modes that depend on the binary's past history. The most well-known of these is the displacement memory effect which is dominant in the $l = 2, m = 0$ mode, and the next leading memory effect, known as the spin memory, is dominant in $l = 3, m = 0$ mode for the non-precessing binaries (see e.g., [327] and [328]). There are other higher-order memory effects, but these can be extremely sub-dominant. Most of these are discussed in [329] and references therein. While these are small effects, they will need to be included to prevent biases, and have now been included in a surrogate model for nonprecessing (quasicircular) binary black holes constructed using the waveforms obtained from Cauchy-characteristic evolution [330]. The effect of non-linear memory on the binary black hole parameter estimation is studied in [331] where the dominant displacement memory in the $l = 2, m = 0$ mode starts to affect the parameter inference at SNR $> 60$ for the current generation ground-based detectors (such as LIGO A$^{\#}$). Moreover, the effect of memory has been studied in the case of neutron star-black hole and binary neutron star mergers [332, 333], where it is argued that the memory can affect parameter estimation for the XG detectors. Studies show that it will be difficult to detect the presence of memory in individual sources with the current LIGO, Virgo, and KAGRA detectors at O4 sensitivity or even O5 sensitivity, but it could be detected in a population using the stacking procedure (e.g., [334]). Thus, it is necessary to understand the effect of memory on parameter estimation and tests of GR at the population-level.

Therefore, analyzing a GW signal that has a significant magnitude of spin-induced precession, higher order mode content, and memory effect with an inaccurate or incomplete waveform model may not only deteriorate parameter estimation, but also show biases in the inference of other source parameters (see, e.g., [312]). A recent study has investigated systematics due to waveform mismodeling by comparing SEOBNRv5PHM and IMRPhenomXPHM. It was found that systematic biases can impact the current and future GW-detector networks, affecting the inference of realistic binary black hole population properties, as well as, the science cases of individual loud signals [248], and more in general binaries with large mass ratios and high precession. Such systematic biases may eventually find their way into the measurement of a beyond-GR parameter depending on the nature of its correlation with the other source parameters, inducing a false violation of GR. Hence, it is essential to use accurate waveform

models with spin-precession effects, *sufficient* number of higher-order modes, and memory effects while testing a GW signal for a violation of GR.

### 3.2.2 Sub-optimal Calibration and Agreement With NR Waveforms

State-of-the-art waveform models are built by combining and resumming information from different analytical methods, such as PN approximation and gravitational self-force theory, and then calibrating/validating against NR simulations and merger-ringdown waveforms in the test-particle limit, which are obtained by solving the Teukolsky equation. The assessment of the accuracy of the waveform models from the two main waveform families (notably EOB and IMRPhenom models) can be found in [248, 315, 320, 322, 335–337]. Due to the number of calibration parameters and the large number of NR simulations at disposal, it is especially important to devise a computationally efficient and flexible calibration procedure. For instance, in calibrating the SEOBNRv5HM model [320], the authors quantified the agreement with NR waveforms in a Bayesian fashion and employed nested sampling to obtain posterior distributions for the calibration parameters. State-of-the-art waveform models use best-fit estimates across the physical parameter space for their calibration parameters. Providing instead a probability distribution, modeled for example through a multidimensional Gaussian mixture, would allow accounting for uncertainty estimates due to sub-optimal fits, and could mitigate waveform systematics at high SNR [338]. Other proposed methods to marginalise over waveform modeling uncertainties include Gaussian process regression [339–342], or introducing frequency-dependent amplitude and phase corrections, as in the case of detector calibration uncertainty [164]. While these methods may obscure small deviations from GR, particularly around the merger phase, significant deviations that exceed the estimated modeling uncertainties should still be detectable.

Calibration parameters typically enter in waveform models as higher-order PN coefficients, which are currently unknown. Including higher-order analytical information, while pushing the calibration parameters at even higher orders, could improve the accuracy of current waveform models, but requires careful studies on how to incorporate and resum this information [320, 335, 343] Nonetheless, neglecting higher-order PN terms carries an error which might become relevant with updates to current detectors and XG detectors, but could be mitigated by marginalizing over higher-order PN coefficients as new model parameters [344]. Incorporating results from the post-Minkowskian (PM) approximation [345–348], a weak fields expansion in $G$ at all orders in the velocity, is also promising, particularly for highly eccentric binaries for which relativistic velocities can be reached at each periastron passage even in the weak field regime. While PM results have not yet been incorporated in state-of-the-art waveform models for bound orbits, remarkable agreement has been obtained comparing PM-improved EOB models to NR for scattering orbits [349–352].

The calibration procedure imposes that the waveform model agrees, as much as possible and for the entire coalescence, with the NR waveform. This is often quantified by computing the unfaithfulness (or mismatch) $\mathcal{M}$ between the model and NR waveform. As detectors become more sensitive and the SNR increases, the accuracy requirements become more stringent, thus demanding smaller unfaithfulness values. Accuracy requirements are usually formulated in terms of an indistinguishability criterion [353–357], which states that if two waveforms fulfill the condition

$$\mathcal{M} < \frac{D}{2\,\text{SNR}^2}, \tag{1}$$

for a given power spectral density (PSD) and SNR, then these waveforms are considered indistinguishable, and differences in the recovered parameters are expected to be smaller than statistical errors. Here $D$ is an unknown coefficient, usually set to the number of intrinsic parameters of the source [356] or tuned with synthetic injections at increasing SNR [357]. Being

sufficient, but not necessary, this criterion is generally too conservative, and, if it is violated, differences are not necessarily measurable, or may appear in a subset of parameters in which one is not typically interested [357, 358]. Toubiana & Gair [359] recently proposed a correction to the standard indistinguishability criterion by revisiting some of the hypotheses under which it is derived, and employed it to quantify apparent deviations from GR due to waveform inaccuracies [360].

The state-of-the-art multipolar, aligned-spin SEOBNRv5HM model shows a median unfaithfulness of $1.01 \times 10^{-3}$ against 442 NR waveforms when using the O5 PSD [361] and considering binary total masses in the range $[20-300]M_\odot$. Using this model would lead to a false deviation from GR when measuring the QNM (complex) frequencies of a massive binary black hole with a mass ratio of 2, as observed in LISA with an SNR $\mathcal{O}(100)$ [360]. This issue occurs because for such massive binary black holes, the majority of the SNR lies in the merger-ringdown stage. By contrast, a stellar-mass binary black hole with mass ratio 6, observable in O5, would not incorrectly lead to a violation of GR at SNR 75 [282], because in this case a large portion of the SNR is accumulated during the inspiral stage. Normally, the accuracy of waveform models gets worse toward merger, where the presence of higher-order modes becomes more and more important, while their modeling is quite challenging. The recent study of [362] investigated the impact of inference biases from sub-optimal waveform calibration on a realistic population of binary black holes in XG detectors. They considered two quasi-circular, nonprecessing waveform models of the same family (namely, IMRPhenomD [363] and IMRPhenomXAS [364]) and estimated a mismatch requirement of $\sim 10^{-5}$ for 99% of the events with SNR $> 100$ not to be biased.

Inaccuracies in NR waveforms, due to, e.g., numerical truncation errors and issues with GW extraction and extrapolation, are typically at least one order of magnitude smaller than errors between semi-analytic models and NR [357]. Nonetheless, they are expected to become relevant with updates to current detectors and XG detectors, especially for binaries with asymmetric masses and orbits inclined with respect to the line of sight [357, 365, 366].

# 4  Astrophysical Aspects

There are several astrophysical aspects of the source, its surroundings, and the emitted GW signal that have not been accounted for in the state-of-the-art waveform models. These aspects, if present in the real GW signal, might affect the tests of GR and can lead to false GR violations. Here we discuss those astrophysical aspects that we can think of.

## 4.1  Gravitational Lensing

As GW detectors get upgraded and new ones join the network, more and more distant mergers can be observed. This increases the chance of having a matter density crossing the GW travel path, possibly leading to gravitational lensing. Depending on the lens properties and the lens-source geometry, different effects can be observed. For the best-aligned and most massive cases, we are in the geometric optics limit and lensing leads to several copies or "images" of the initial signal. These images have the same frequency evolution but are delayed in time, (de)magnified, and can undergo an overall phase shift. When the time delay is large enough, these images are distinct, and we face strong lensing [367, 368]. For ground-based detectors, typical lenses are galaxies and galaxy clusters [369]. For smaller time delays, corresponding to less aligned systems and lighter lenses, one has millilensing, where the various images overlap and sum to a non-trivial signal in-band [370]. This is expected to be due to heavy black holes, or dark matter over-densities, for example. Finally, when the GW wavelength is comparable to or greater than the size of the lens, we need to perform the full wave-

optics treatment [367], and lensing leads to frequency-dependent beating patterns known as microlensing. For ground-based detectors, typical lens sources are individual stars, black holes, or dark-matter overdensities [371]. It is also important to note there can be interplay between these different types of lensing. When strong lensing happens, one or more of the images may undergo micro or millilensing because of individual objects present in the strong lens [372–374].

False GR deviations could be expected when GR signals are distorted. For strong lensing, one can have such an effect for specific values of the overall phase shift. In particular, it can take only three distinct values: $0$, $\pi/2$, or $\pi$, corresponding to a minimum, saddle point, or maximum of the Fermat potential, and referred to as Type I, II, and III images, respectively [368, 375]. Under all circumstances, Type I and III images are indistinguishable for the GR case because they correspond to no shift or a sign flip in the polarization, which cannot be detected [375]. For Type II images, on the other hand, detectability is possible when the GW displays higher-order modes. In this case, the phase has different pre-factors for different frequency modes and is not degenerate with the (frequency independent) lensing phase shift anymore [375]. This can be used to detect strong lensing based on a single image, although it requires rather large SNRs and very asymmetric, precessing or eccentric systems [375–378]. When analyzing Type II images under the unlensed assumptions, one can face losses in SNR, possibly missing the event with template searches [376], or biases in parameter estimation [377, 378]. Therefore, one can expect this non-trivial feature to also be picked up when searching for GR deviations. For example, this is the case with modified dispersion relations that change the frequency evolution of the GW phase in a way possibly similar to lensing [379]. The link between Type II images and GR deviations is also highlighted in [380], where the authors show that some GR deviations are flagged by Type II search pipelines.

The cases of millilensing and microlensing are even more favorable in leading to spurious GR deviations being detected since they both lead to a non-trivial signal in the detection band, although the nature of the resulting image is different between the two cases [367, 370, 371]. When analyzing such signals with traditional GR templates, one expects imperfect modeling of the signal, leading to coherent power left in the data [381]. This is also confirmed in [382] for some tests of GR. In this study, the authors show that milli and microlensed signals can lead to spurious deviations from GR, sometimes with a high significance. However, it is also important to note that adapted lensing pipelines also clearly see these events as being lensed. Therefore, the GR deviation would probably not be confirmed as it would be explained via lensing, underlying the importance of accounting for possible astrophysical effects on the GW signals when looking for GR deviations. The link between GR deviations and micro and millilensing is also further confirmed in [380], where the authors show that some deviations of GR lead to false positives in micro and millilensing searches. In the case of a multi-messenger lensing event in which the GW lensed signal is in the wave optics regime but the electromagnetic signal is in geometric optics (which is to be expected given their higher frequency), the speed of propagation of GWs could appear to be superluminal due to the waveform distortions [383], although no information actually arrives faster than light [384].

A crucial approximation in these studies is the exclusion of the effect of parallel-transporting the polarization tensor across the lensing geometry and the treatment of GWs as scalar waves which become increasingly violated as one moves from the weak gravity limit. Recent studies [385, 386] have pointed out the consequences of such an approximation and started treating GWs as a tensor field. It is pointed out that there is no notion of a unique "propagation direction" as can be defined in the geometric optics limit as well as the wave optics treatment for a scalar wave. Similarly, strong gravity effects could add extra phenomenology [387].

Therefore, all types of lensing—micro, milli, and strong—can potentially lead to spurious GR deviations being detected if neglected. Hence, should such deviations be seen, it would

be crucial to verify possible astrophysical origins of the modification in the GW signal, and in particular if the GW event is not lensed.

## 4.2  Environmental Effects

The current waveform models can be referred to as *vacuum templates* as they only describe GWs from isolated binary systems in a vacuum environment, neglecting realistic astrophysical surroundings of the source. However, in reality, the binary is always in an astrophysical environment that impacts the binary's orbital evolution and hence results in a GW signal from the binary different than the vacuum template. There are many scenarios in which the GW signal from an environment-embedded binary system could be different from its corresponding vacuum signal. These are, but not limited to, (i) the source resides in a dense environment [388–391] such as dense cores of massive stars [392–394], accretion disks of active galactic nuclei [32, 395–400], and star clusters (see, e.g., [401]), (ii) the source resides in a dark matter halo [32, 402–407], and (iii) the source is immersed in a strong electromagnetic field [408, 409]. Moreover, the peculiar acceleration of the source with respect to the observer, i.e., time-varying Doppler shift [410–413] and the acceleration of the universe, i.e., time-varying redshift [410, 414, 415] itself could lead to GW signals being different from vacuum templates.

The situation where there is a massive bosonic field that is amplified around black holes via superradiance (see, e.g., [416]) is also sometimes considered an environmental effect and can similarly lead to deviations from a vacuum binary black hole signal. However, there are significant differences in this case compared to the environmental effects considered above. Most importantly, in this case the size of the deviation is set by the universal properties of the boson and the properties of the binary, not the specifics of where the binary formed. This makes the deviations more similar to a deviation from GR (which also depends on the properties of the binary and some universal parameters). However, there are many ways to distinguish binaries of black holes with boson clouds from GR deviations. Some of these are discussed in Sec. 4.3.2, since the emitted GW signal in such a scenario will be similar to the one from binaries of black hole mimickers (e.g., there will be tidal effects from the boson clouds). Additionally, since the superradiant growth of the clouds is only possible for certain pairs of black hole masses and spins (see, e.g., [417]), it should be easy to distinguish this case from modified gravity (or black hole mimickers) when considering the population. The time dependence of the tidal deformability and non-black hole multipole moments due to perturbations or even disruption of the clouds due to the effects of the other black hole (see, e.g., [418–420]) should allow one to distinguish the boson cloud case from black hole mimickers even for individual sources. Additionally, one can obtain constraints on the boson mass from the contributions from the superradiant instability to the stochastic background of GWs [421, 422]. Furthermore, boson clouds are expected to emit a nearly periodic and long-duration GW signal [421, 422] and no evidence of such signals is found in current GW data, which provides constraints on the ultra-light scalar boson field mass (see, e.g., [423–426]).

Returning to environmental effects proper, the detailed modeling of different environmental effects on the binary's GW signal is challenging and requires computationally expensive NR simulations [393]. However, in the literature, these effects have been approximated as a correction to the vacuum GW signal's PN phase evolution. For example, at the leading order, dynamical friction due to gas accretion can be modeled as a $-5.5$PN correction whereas collisionless (collisional) accretion can be modeled as a $-4.5$PN ($-5.5$PN) correction [391, 427–429]. The accretion and dynamical friction due to a scalar dark matter cloud give rise to a $-4$PN and $-5.5$PN correction, respectively, to the phase at the leading order [430]. Electromagnetic effects have been computed at next-to-leading order (at 3PN) by taking into account the whole electromagnetic structure of a star. The leading magnetic corrections at 2PN order (assuming

a constant and aligned magnetic dipole) to the GW phase are found to be comparable to a 1.5PN point-particle effect [431, 432]. Phase correction due to the line-of-sight peculiar acceleration of the source has been computed up to 3.5PN order [411, 433] while the acceleration of the universe leads to a −4PN correction to the phase at leading order [414, 415].

It has been argued that the magnitude of the environmental [32, 434] and cosmological [410] effects are expected to be quite small and hence could be neglected for ground-based detectors. However, there could be scenarios where these effects are non-negligible, e.g., stellar-mass compact binaries would merge around a supermassive black hole and one can still get a significant deviation from the vacuum template in the bands of LIGO/Virgo/KAGRA detectors [433]. Moreover, near supermassive black holes, in galactic nuclei, triple systems of stars are common and they mostly are hierarchical in nature [435–437], i.e., a tight inner binary is orbiting a tertiary on a wider orbit which forms the outer binary. In these *hierarchical triples*, the tertiary brings interesting features to the GW signal emitted by the inner binary, e.g., the oscillation of eccentricity and inclination of the inner binary's orbit due to the Kozai-Lidov mechanism [438, 439]. Such oscillations could modify the frequency evolution of the inner binary and this needs to be taken into account in waveform modeling [440, 441].

A recent study by Santoro *et al.* [442] showed that particularly large environmental effects can significantly bias the parameter estimation if vacuum templates are used for the analysis, even when not directly detectable by LIGO-like instruments. Although this bias requires extremely dense environments that are not predicted by standard astrophysical models, it would be important to find out if such biases in parameters could lead to false GR violations for more sensitive XG detectors.

Likewise, ringdown templates are simple and based on predictions from vacuum GR. Modifications of GR usually lead to extra polarizations or include degrees of freedom with different modes, introducing a simple handle to test for beyond-GR physics. However, environmental effects, such as accretion disks, dark matter halos or any form of matter outside of black holes introduces low-frequency modes or drastic changes to higher overtones, de-stabilizing the spectrum [32, 443, 444]. Concrete examples suggest that spectral instability of the dominant mode introduces changes in the waveform only well after coalescence, but the relevance of overtone instability for time-domain waveforms still needs to be well understood [445].

However, it is worth mentioning that environmental effects will be possibly important only for certain events, while likely negligible for the majority. Thus, any competing beyond-GR interpretation of environmental effects should coherently explain this non-trivial dependence on the source.

## 4.3 Mistaken Source Class

### 4.3.1 Beyond Compact Object Mergers on Bound Orbits

Parabolic or hyperbolic scattering [446] as well as head-on collision of compact objects [447–449] may give rise to GW signals which may resemble that of a quasi-circular CBC close to the peak of the signal. Therefore, for relatively short-duration signals, there is a risk of confusing a compact binary merger with one of the above classes of sources, leading to biases on the source parameters and thereby affecting tests of GR. In the case of GW190521, studies have discussed the degeneracy between a precessing compact binary in quasi-circular orbit with a binary that undergoes head-on collision [450] and a merger of two nonspinning black holes on hyperbolic orbits [451]. It is argued that the lack of premerger features in certain precessing configurations in quasi-circular CBC may mimic a head-on collision leading to underestimation of mass parameters and overestimation of luminosity distance when a quasi-circular CBC waveform is employed for parameter estimation. Obviously, such biases will directly affect most tests of GR.

However, precise estimates of final spin can help in distinguishing head-on collision from a quasi-circular CBC. For example, if the inferred remnant black hole spin is high (e.g., $\sim 0.7$ as was the case for GW190521), this could make the head-on collision unlikely as very special configurations may need to be invoked to explain this. As the head-on collisions are themselves very special configurations, additional requirements such as this (large remnant spin) may weigh down their possibility in a model selection problem. Further, due to the special symmetries of the head-on collision, the spherical harmonic modes excited in a head-on collision may differ from those in a quasi-circular CBC. For instance, unlike quasi-circular CBCs, in head-on collisions $\ell = 2, m = 0$ mode may be as strong as $\ell = 2, m = 2$. Such features may also help in a model selection problem. A dedicated study that looks into the effect of degeneracy between quasi-circular CBC and head-on collision or parabolic/hyperbolic encounters and how that impacts tests of GR will be very useful. To do this we require more accurate analytical or numerical waveform modeling of head-on collision and parabolic/hyperbolic encounters.

### 4.3.2 Black Hole Mimickers

There are various exotic compact objects that are massive and compact enough that gravitational waveforms from binaries of such objects could be close to those from a binary black hole (see, e.g., [452, 453]). The simplest such objects can be described by GR minimally coupled to a non-Standard Model field (e.g., an ultralight scalar field describing dark matter [454]). More complicated models for such objects involve nonminimally coupled fields, where it may make more sense to treat the additional scalar field as part of the gravity sector. However, even in the case where gravity is still GR, the specifics of the waveform would still differ from that of a binary black hole in GR, and one would thus obtain a false deviation from GR when applying a test of GR based on a binary black hole waveform model. The most theoretically well-modelled such objects are boson stars (see, e.g., [455]), which are formed from a massive complex scalar or vector field, that may be self-interacting, as is necessary to obtain more compact stars (that are thus more similar to black holes)—see, e.g., [456]. However, there are many other models, including quite exotic objects, like gravastars [457], which have an interior made of de Sitter space. A concrete framework for these exotic objects might require GR deviations [458], but they can be described also using exotic matter within GR (e.g., [459]).

For all of these cases, there will be the same matter effects on the inspiral that one finds in the PN approximation for binary neutron stars (some of which are discussed in Section 3.1.2), albeit with different values. In particular, there will be effects of nonzero tidal deformabilities (see, e.g., [456, 459–461]), tidal disruption (at least for sufficiently unequal-mass binaries; see, e.g., [462]), and the excitation of resonant modes in the objects (see, e.g., [463]), as well as effects from multipoles that are different from those in black holes (see, e.g., [173, 464]) and possibly a lack of the relatively large GW absorption (a.k.a. tidal heating) one obtains with black holes (see, e.g., [465]). However, since recent studies [206, 466] have shown that neutron stars can have larger GW absorption than black holes if they have a sufficiently large shear viscosity, it is possible that the same is true for some potential black hole mimickers, though this is not the case for, e.g., standard models of boson stars. There will also be differences in the merger-ringdown part of the signal (see, e.g., for simulations of orbiting binary boson stars [467–470]). If the merger of a binary of exotic compact objects forms an ultracompact object (i.e., an object that has a light ring outside its surface), then the ringdown is nearly indistinguishable from that of a black hole and a train of modulated pulses—known as GW echoes—is emitted in the late postmerger stage [32, 471]. From the analysis of current GW events, no evidence for postmerger echoes has been found with unmodelled and modelled searches [4, 5, 472–477], despite claims of echo detections in [478–481]. Moreover, for perfectly reflecting objects the presence of echoes is disfavored by the current upper bounds on the stochastic background in the advanced LIGO frequency band [482].

If one has a single population of exotic stars that are formed from a single fundamental field, then the non-GR effects in the inspiral will be solely determined by the masses of the objects, and there will be a maximum mass of stable stars, just as in the neutron star case. Thus, if one can measure these effects (and the masses of the stars) accurately (using, e.g., a more refined version of the analysis given in [483]), then one can check if the signals are indeed consistent with coming from a population of binaries of such stars. While alternate theories of gravity with an intrinsic scale will have a roughly similar behavior, where the GR deviation decreases with increasing mass of the black holes, it seems unlikely that an alternative theory of gravity would be able to mimic the situation of exotic stars to a high degree of accuracy. Moreover, if there is a population of exotic binaries as well as binary black holes, then one may observe binary black holes with very similar masses, spins, and distances as the exotic binaries, where a modified theory would predict that one would also observe deviations for the black hole binaries. Thus, while it is likely that the two situations could be confused with initial observations, it should be straightforward to distinguish them with high-accuracy observations. However, the ability of a given set of observations to distinguish specific exotic star models and specific alternative theories would need to be tested with explicit calculations.

For instance, black holes can have nonzero tidal deformabilities in certain alternative theories, such as those that introduce higher-order-in-curvature corrections in the action [460, 484]. However, in such models the dimensionless tidal deformabilities are proportional to inverse powers of the black hole mass, $1/M^n$, where $n$ is a positive integer that depends on the theory ($n = 4$ or $6$ in the calculations cited). This is not a good match for the mass dependence of any of the boson star models considered in [460], and while it might be possible to find an exotic star model that gives a better match, the stars would still have a maximum mass, while the black holes in the alternative theory have nonzero tidal deformabilities for all masses. The black holes also have differences in the spin-induced multipoles (see, e.g., [485]) that would also have to be reproduced by the exotic stars, which is unlikely to be possible to more than moderate accuracy. For instance, for some families of boson stars, the spin-induced moments have minimum values larger than their Kerr values (similar to the minimum values of tidal deformability), and show a different spin dependence than one obtains for alternative theories (see, e.g., [486]). Additionally, there will be differences in the GW absorption comparing black holes in this theory and black hole mimickers with no horizon (which will generally have a much smaller GW absorption cross section than black holes). However, one also expects that the GW absorption in such theories will differ from that in GR due to the differences in the static tidal response, given the relation between this and GW absorption/tidal heating (see, e.g., [487]). Moreover, there are also changes to the binary's dynamics that do not come from finite size effects in such theories (see, e.g., [488]), albeit only occurring at high PN orders.

Thus, individual signals from binaries of exotic compact objects could be confused with a GR deviation in many tests (which do not include the expected non-black hole modifications to the waveform). However, binaries of black hole mimickers will in general be able to be distinguished from a modification to GR, even one that predicts nonzero tidal deformabilities for black holes, at sufficiently high SNRs and when analyzing the population of signals, or possibly when performing multiple independent tests of a single signal.

## 4.4 Statistical Assumptions of Astrophysical Population

Combining information from multiple signals is a powerful method to perform stronger tests of GR. However, assumptions on the underlying astrophysical population and the statistical methods adopted to perform the joint analysis can affect the results.

Biases due to waveform modelling systematics can pile up when stacking multiple events in a catalog. Several studies [90,136,489,490] show that even if systematics are under control at the level of the individual events, the accumulation of biases in a population analysis can

produce false deviations from GR if the catalog is large enough. Depending on the actual population of resolved signals and on the way the events are combined, false deviations can appear with as little as $\sim 30$ events with SNR$> 20$ in the most pessimistic scenarios [490]. Moreover, restricting the study to golden events with high SNR is even more vulnerable to false deviations once these events become routine in XG detectors [90, 136], although techniques to mitigate the biases have been proposed [489].

Furthermore, combining events requires concrete assumptions about the impact of the astrophysical population and the detectability of GW sources that violate GR. Many parameterized tests of GR infer the presence of expected correlations between individual source parameters (such as the total mass of a binary black hole system) and the deviation parameter [491]. These correlated features within the inferred posterior distributions for individual events imply that specific choices regarding the astrophysical population distribution can skew these results to different regions of the parameter space.

In a recent study, Payne *et al.* [492] demonstrate that neglecting the astrophysical population leads to inferences which are $\sim 0.4\sigma$ less consistent with GR within GWTC-3 for parameterized tests of GR. However, they show that such biases can be mitigated by jointly inferring the astrophysical population properties while combining the distributions of GR violation parameters. Furthermore, Magee *et al.* [493] illustrate that neglecting the loss in detectability of signals with GR violations places constraints on PN deviations that are up to 10% too narrow when ignoring the selection bias in the population. These studies highlight the need to carefully consider the underlying statistical methodologies used when attempting to test GR. In the same vein, astrophysical inaccuracies or biases in the properties of a source population (e.g., imperfect mass distributions) could also lead to false GR deviations. For example, this can happen if events are detected in regions of the parameter space disfavored by astrophysical population models, such as hierarchical mergers from black holes residing in the theorized upper-mass gap [494]. Tests of GR will need to adopt the ever-growing astrophysical population knowledge to remain sufficiently unbiased.

Combining events to test GR also requires assumptions on the GR deviations that are being tested. If the GR modification is common among all the events (as in the case of, e.g., a nonzero graviton mass or a nonzero time variation $\dot{G}$ of Newton's constant), one can multiply the individual, marginalized likelihoods on the deviation parameter to obtain the combined likelihood for the catalog [90, 490, 495, 496]. On the other hand, if the GR deviations are independent for each event (as may be the case if black holes have "hair"), one can multiply the individual Bayes factors in favor of GR to obtain the total evidence from the catalog [90, 490, 495]. In a more general framework where the distribution of GR deviations across the catalog is a known function of the event parameters (such as masses, spins, and compactness), one would need to perform a full Bayesian hierarchical inference on the population [495, 497].

Studies have shown that testing GR at the population level under one of the three assumptions listed above (that all events share the same beyond-GR parameter; that modified theories introduce a new unrelated parameter for each detection; or that GR deviations across the catalog are a known function of the event parameters) can lead to the wrong conclusions if the underlying GR deviation does not satisfy the assumption [495, 497]. Moreover, the accumulation of biases across the catalog due to waveform systematics can change significantly depending on which method is chosen to combine multiple events [90, 490]. Recent work by [498] suggests that performing a full Bayesian analysis should be the most robust approach, but it still requires assumptions that can make the inference inherently model-dependent [495].

As shown by [499], the finite size of the observed catalog will produce cosmic-variance effects that can cause to incorrectly infer deviations from GR, but a bootstrapping technique can be used to mitigate this effect.

## 5    When Does a Cause Become Important?

Not all effects discussed in this paper are created equal, with some being always important for understanding false GR violations, such as non-stationary noise artifacts and glitches (see Sections 2.1 and 2.2) while some will not be important until XG detectors or beyond, such as unaccounted effects of the physics of gas and dust in the environment of binary black hole mergers (see Section 4.2). In this Section, we gauge when each of these causes will become important in terms of the generation of GW observatory.

It is worth stressing that some level of systematics is unavoidable. For example, waveform models are intrinsically imperfect: even without missing any physics and removing current waveform systematics, there will always be intrinsic limitations due to truncation errors in perturbative schemes, calibration inaccuracy with NR waveforms, phenomenological modelling of the merger, unavoidable numerical errors in NR simulations. Thus, we will have to always face some degree of waveform systematics, noise artifact, or astrophysical uncertainty, whose potential impact will grow for high SNR events. The point here is to control such systematics as much as possible, to a level that make them negligible with respect to a putative GR deviation.

We summarize the discussion in Table 1. We note that this is intended as a rough guide as exact predictions for the size of relative effects can depend on a number of factors, and one expects improvements in the coming years (e.g., one expects waveform systematics to improve in the coming years, however, we do not consider this here). Below we give our reasoning for why we think these causes will be important (or not) for a given detector sensitivity.

| Cause | O4 | A+ | A$^{\#}$ | XG |
|---|---|---|---|---|
| Non-Stationary Noise | ✓ | ✓ | ✓ | ✓ |
| Non-Gaussian Noise/Glitches | ✓ | ✓ | ✓ | ✓ |
| Overlapping Signals | ✗ | ✗ | ✗ | ✓ |
| Data Gaps | ✗ | ✗ | ✗ | ✓ |
| Detector Calibration | ✗ | ✗ | ✗ | ✓ |
| Eccentricity | ✓ | ✓ | ✓ | ✓ |
| Tidal Effects | ✗ | ✓ | ✓ | ✓ |
| Kick-induced Effects | ✗ | ✗ | ✗ | ✓ |
| Ringdown Modes | ✓ | ✓ | ✓ | ✓ |
| Precession and Higher-order Modes | ✓ | ✓ | ✓ | ✓ |
| Memory | ✗ | ✗ | ✓ | ✓ |
| Sub-optimal Waveform Calibration | ✗ | ✗ | ✓ | ✓ |
| Lensing | ✗ | ✗ | ✗ | ✓ |
| Environmental Effects | ✗ | ✗ | ✗ | ✓ |
| Source Misclassification | ✓ | ✓ | ✓ | ✓ |
| Astrophysical Population Assumptions | ✓ | ✓ | ✓ | ✓ |

Table 1: Summary of the causes discussed in this paper that can potentially mimic a GR deviation while performing tests of GR. The tick means the effect should be accounted for in the waveform models and/or analysis methods when analyzing data of a GW detector of a given sensitivity. The cross means the effect is sub-dominant to show up as a false GR violation with that detector sensitivity.

### 5.1    Noise Systematics

**Non-stationarities, non-Gaussianities, overlapping signals**    Non-stationary and non-Gaussian noise artifacts are an ever-present analysis burden in the current generation of observatories

as discussed in Sections 2.1 and 2.2. While the extent to which these artifacts will alter with upgrades to current observatories or persist in future-generation observatories remains uncertain, it is difficult to imagine that they will subside to any degree. It therefore behooves analysts to understand and mitigate these noise sources as post-processing steps before making any claim of a GR violation. On the other hand, the effect of contamination from overlapping signals, whether they be super- or sub-threshold to detection, will only increase and get worse as the sensitivity of instruments gets better.

**Data Gaps**    For current-generation detectors, data gaps are not expected to pose a problem for tests of GR due to the typically short duration of signals in the band and the low likelihood of data gaps occurring precisely during those times. For XG observatories, however, data gaps could become more problematic, as the signal duration increases to many hours to days, and the likelihood of gaps increases.

**Detector calibration**    For the current generation of observatories, uncertainties due to detector calibration do not introduce biases in parameter estimation when assuming general-relativistic waveforms, and therefore are not expected to introduce problems in tests of GR (e.g., [102] and see Section 2.5). For XG observatories, assuming an $\approx 1\%$ relative error on the amplitude, and $\approx 1°$ error in phase, detector calibration error leads to mismatch errors of approximately $10^{-5}$, which may be problematic for tests of GR [500]. Of course, this is only a dominant source of uncertainty if other sources (e.g., waveform systematics) can be mitigated below this level.

## 5.2   Waveform Systematics

**Eccentricity**    Employing non-precessing, eccentric waveforms, some papers have claimed the evidence for eccentricity in observed GW signals [128, 129, 138–140]. Although this is contentious (see discussion in Section 3.1.1), it points to the fact that effects of eccentricity are already relevant in current observations, and therefore already pose a difficulty when performing tests of GR. This will continue to be a problem, and may be further exacerbated, as observatories become more sensitive.

**Tidal Effects**    Tidal signatures may be present in several observed neutron star binary mergers (e.g., [62, 501]), although a confident detection of tidal signature is yet to occur. While misspecification of tidal effects is unlikely to appear as a GR violation in current detectors, a clean tidal signature may be present in A+ observatories for dynamical tidal effects [502], and XG detectors for linear tides (e.g., [503, 504]).

**Kick-induced Effects**    The kick-induced effects are too small to be detected with the current GW detectors but could potentially be observed in XG era [234, 235]. The XG detectors are expected to observe $\sim 4-5$ events per year for which these effects will be constrained to better than $\sim 10\%$ [234].

**Ringdown**    Tests of GR and the no-hair theorem are already performed using the ringdown of loud GW signals (e.g., [1]) where the challenges that arise with specifying the ringdown start time and avoiding overfitting to nonlinearities are already present. These challenges will only intensify as the ringdown signals become louder in future observatories (e.g., [286]).

**Precession and Higher-order Modes**   Several events in the existing GWTC have strong evidence of higher-order modes due, e.g., to large mass ratios such as GW190412 [505] and GW190814 [506]. There are several events that have evidence of spin precession, such as GW190521 [507] and GW200129 ( [70], although see [71,508]). It is therefore important to account for spin precession and higher-order modes in current analyses, and the inclusion of higher modes will become even more important as the sensitivity of observatories continues to improve.

**Memory**   Displacement memory is too small to be detected in individual events with the sensitivities of current detectors [334, 509–511]. A memory signal is expected to influence parameter estimation results in loud events with SNR greater than 60, expected during the $A^{\#}$ era [331], implying at this stage memory needs to be properly accounted for in waveforms models. Memory will have a significant influence in XG observatories; for example, Cosmic Explorer is predicted to have 3 to 4 events per year where memory is detectable for an individual event [334], amplifying the need to properly account for memory effects.

**Waveform Calibration**   If we consider NR simulations to be the ground truth, then current waveform calibration errors refer to systematic biases introduced because the waveform approximants do not exactly match the NR simulations. But even NR waveforms carry uncertainties associated with, e.g., resolution effects and finite radius extraction. Such waveform calibration errors on the order of a few percent in amplitude, and a couple of degrees in phase, are subdominant to stochastic noise processes for binary neutron star observations at approximately 100 Mpc in A+ observatories [164]. Waveform uncertaintes are currently smaller than this, implying they are not a potential source of bias for tests of GR. This is not necessarily true in the $A^{\#}$ and XG era when even NR waveforms will not be sufficiently accurate for unbiased parameter estimation recovery [337,500]. This latter point motivates the continual need for more accurate NR simulations and waveform extraction methods, as well as waveform approximations.

## 5.3   Astrophysical Aspects

**Lensing**   In current and future detectors like advanced LIGO and A+, the estimated rate of strong lensing events for binary neutron stars is approximately 0.1%, while for binary black holes it is expected to be around 0.2%. These figures are consistent across various studies [512–514]. Following this, advanced LIGO is anticipated to detect approximately 0.1 lensing events per year, whereas A+ is projected to observe 1 event annually. However, with XG detectors, $\mathcal{O}(100)$ events could be detected per year. It is important to note that these rates serve as a lower bound for millilensing and microlensing, since they could occur together with strong lensing in events. Therefore, lensing effects will not be a significant issue only until XG era.

**Environmental Effects**   Astrophysical environments in which one may anticipate binary systems merging (and which may leave an imprint on the GW signal) include thick ($\bar{\rho} \sim 10^{-8}$ g/cm$^3$) and thin ($\bar{\rho} \sim 0.1$ g/cm$^3$) accretion disks around active galactic nuclei [32], cold dark matter spikes ($\bar{\rho} \sim 10^{-6}$ g/cm$^3$) [404], superradiant-boson clouds ($\bar{\rho} \sim 0.1$ g/cm$^3$) [416] and the dynamical fragmentation of massive stars ($\bar{\rho} \sim 10^7$ g/cm$^3$) [393]. Santoro *et al.* [442] found no support for environmental effects in GWTC-1, and found the environmental density would need to be $\sim 20$ g/cm$^3$ to be observable. This likely does not correspond to any of the astrophysical environments mentioned previously. For advanced LIGO design sensitivity, they find that dynamical friction effects are detectable at $\bar{\rho} \gtrsim 10$ g/cm$^3$ for a GW170817-like event,

while the effect of collisionless accretion is only visible for densities 10-100 times greater. As there are no proposed environments with such densities, it is unlikely for environmental effects to be visible in advanced LIGO data. They find that XG observatories will be sensitive to environmental densities of $\sim 10^{-3}$ g/cm$^3$, which includes both thin accretion disks and super-radiant clouds. It is therefore likely that environmental signatures will only become relevant for GR tests in XG and beyond.

**Source Misclassification**    The problem of source misclassification is ever-present in tests of GR and must be considered when mitigating against false GR violations. For example, while current analyses find no evidence of GW echoes that would provide evidence of black-hole mimickers (see Section 4.3.1), these non-detections only place limits on, e.g., the reflective properties of the ultra-compact objects. As the sensitivity of the GW network improves, we will continue to probe the parameter space of potential black-hole mimickers.

**Astrophysical Population Assumptions**    The problem of fortifying hierarchical tests of GR against population assumptions and modelling systematics will be ever-present. Statistical assumptions on how to combine the information from individual events require care, as they reflect implicit assumptions on the beyond-GR theory that is being tested [241, 495]. Incorrect prior assumptions on the astrophysical population can cause biases if the deviation parameters are correlated with individual source parameters. These biases can be mitigated by jointly inferring the astrophysical population when performing hierarchical tests of GR, or in the high-SNR limit of XG detectors if the degeneracies between source parameters and deviation parameters are not perfect [492]. Effects due to the finite size of the catalog [499] or selection effects against large deviations [493] can also lead to biases in population constraints if not properly accounted for. Finally, waveform systematics (both due to missing physics and sub-optimal calibration) can accumulate in a population analysis and lead to infer false GR violations even if the biases are under control at the single-event level [136, 490]. This effect will be even more prominent when restricting the test to high-SNR events that can be routinely observed with XG detectors [90].

# 6   Summary

Since the first detection in 2015, GW observations are now routinely used to test GR in highly dynamical and non-linear gravity regimes. Several tests of GR exist at the moment and the majority of them rely on comparing the GW data with well-motivated, state-of-the-art wave-form models. The GW observations from the LIGO-Virgo-KAGRA collaboration have so far not found any deviation from GR, but this may not be the case forever, especially with the increased sensitivity of GW detectors. In the future, all these well-motivated, state-of-the-art waveform models may fall short of explaining all the features in the high-quality data due to the complexity of the physics of GW sources and the detector noise modeling.

In this paper, we listed the possible causes that can lead to an apparent GR deviation using observations from ground-based GW detectors given the current waveform models and data analysis techniques that are available to the community. We grouped these causes into three broad categories: noise systematics, waveform systematics, and astrophysical aspects. Noise systematics include noise being non-stationary and/or non-Gaussian with or without time-overlapping signals present in the data, gaps in data, and errors in instrument calibration. Waveform systematics include cases of missing physics such as eccentricity, tides, kicks, overtones, mirror modes, and non-linear ringdown modes, and sub-optimal modeling and calibration (with NR waveforms) of quasi-circular waveforms. Astrophysical aspects include

gravitational lensing, non-vacuum environments, mistaken source classes, and assumptions of astrophysical population.

Our list is admittedly not complete and we might have missed some other important causes of false GR deviation. However, we hope that this paper will serve as a starting point for the community to study, understand, and document the effects of these causes on tests of GR. In a follow-up paper, we will discuss what actions could be taken when a significant GR deviation is detected and propose a possible formulation of a GR violation detection checklist. We hope that these efforts will prepare us for the time when there will be an actual statistically significant GR deviation found in the GW data.

# Acknowledgements

**Funding information**  A. Gupta, N.K. Johnson-McDaniel, and P. Narayan are supported by NSF Grants No. AST-2205920 and PHY-2308887. K.G. Arun acknowledges support from the Department of Science and Technology and Science and Engineering Research Board (SERB) of India via the Swarnajayanti Fellowship Grant DST/SJF/PSA-01/2017-18 and the Core Research Grant CRG/2021/004565. K.G. Arun and B.S. Sathyaprakash acknowledge the support of the Indo-US Science and Technology Forum through the Indo-US Centre for Gravitational-Physics and Astronomy, grant IUSSTF/JC-142/2019. E. Barausse acknowledges support from the European Union's H2020 ERC Consolidator Grant "GRavity from Astrophysical to Microscopic Scales" (Grant No. GRAMS-815673), the PRIN 2022 grant "GUVIRP - Gravity tests in the UltraViolet and InfraRed with Pulsar timing", and the EU Horizon 2020 Research and Innovation Programme under the Marie Sklodowska-Curie Grant Agreement No. 101007855. L. Bernard acknowledges financial support from the ANR PRoGRAM project, grant ANR-21-CE31-0003-001 and the EU Horizon 2020 Research and Innovation Programme under the Marie Sklodowska-Curie Grant Agreement no. 101007855. E. Berti and L. Reali are supported by NSF Grants No. AST-2006538, PHY-2207502, PHY-090003 and PHY-20043, by NASA Grants No. 20-LPS20-0011 and 21-ATP21-0010, by the John Templeton Foundation Grant 62840, by the Simons Foundation, and by the Italian Ministry of Foreign Affairs and International Cooperation grant No. PGR01167. V. Cardoso is a Villum Investigator and a DNRF Chair, supported by the VILLUM Foundation (grant no. VIL37766) and the DNRF Chair program (grant no. DNRF162) by the Danish National Research Foundation. V. Cardoso acknowledges financial support provided under the European Union's H2020 ERC Advanced Grant "Black holes: gravitational engines of discovery" grant agreement no. Gravitas–101052587. Views and opinions expressed are however those of the author only and do not necessarily reflect those of the European Union or the European Research Council. Neither the European Union nor the granting authority can be held responsible for them. V. Cardoso has received funding from the European Union's Horizon 2020 research and innovation programme under the Marie Skłodowska-Curie grant agreement No. 101007855 and No. 101131233. S.Y. Cheung, T. Clarke, N. Guttman, P.D. Lasky, L. Passenger, and H. Tong are supported by Australian Research Council (ARC) Centre of Excellence for Gravitational-Wave Discovery CE170100004 and CE230100016, Discovery Projects DP220101610 and DP230103088, and LIEF LE210100002. S. Datta acknowledges support from UVA Arts and Sciences Rising Scholars Fellowship. A. Dhani, I. Gupta, R. Kashyap and B.S. Sathyaprakash were supported in part by NSF Grants No. PHY-2207638, AST-2307147, PHY-2308886, and PHYS-2309064. B. Sathyaprakash also thanks the Aspen Center for Physics (ACP) for hospitality during the summer of 2022. J.M. Ezquiaga is supported by the European Union's Horizon 2020 research and innovation program under the Marie Sklodowska-Curie grant agreement No. 847523 INTERACTIONS, and by VILLUM FONDEN (grant no. 53101 and 37766). E. Mag-

gio acknowledges funding from the Deutsche Forschungsgemeinschaft (DFG) - project number: 386119226. E. Maggio is supported by the European Union's Horizon Europe research and innovation programme under the Marie Skłodowska-Curie grant agreement No. 101107586. A. Maselli acknowledges financial support from MUR PRIN Grants No. 2022-Z9X4XS and 2020KB33TP. S. Tiwari is supported by the Swiss National Science Foundation Ambizione grant no. PZ00P2-202204. K. Yagi is supported by NSF Grant PHY-2207349, PHY-2309066, PHYS-2339969, and the Owens Family Foundation. N. Yunes is supported by the Simons Foundation through Award No. 896696, the NSF Grant No. PHY-2207650, and the NASA Award No. 80NSSC22K0806.

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
