# Peer review of "Possible Causes of False General Relativity Violations in Gravitational Wave Observations"

_SciPost Physics Community Reports, doi:SciPost Phys. Comm. Rep. 5 (2025)_

## Round 1 · Referee Report · Anonymous (Referee 1) · 2024-9-16

Dear Editor,

This is the referee report for the draft titled "Possible Causes of False General Relativity Violations in Gravitational Wave Observations" by A. Gupta et al.

In this paper, the authors conduct a thorough investigation into potential sources of false violations of General Relativity (GR) in gravitational wave (GW) data. They offer a comprehensive discussion of possible contributing factors, which they categorize into three main areas: noise artifacts, waveform systematics, and astrophysical effects. A detailed checklist for identifying strong candidates of GR violation is deferred to a forthcoming publication.

The manuscript is well-structured, clearly written, and accessible. Each potential cause is explained in sufficient detail, allowing even non-experts to grasp the key concepts. The conclusions appear robust and could be of significant interest to the broader community.

I am pleased to recommend this paper for publication in SciPost Physics Community Reports, with some minor comments attached below that the authors might consider addressing to further enhance the paper.

- In Sec. 2.1, it would be helpful if the authors could include examples of both instrumental and environmental sources of non-stationarity. Additionally, they mention that these effects should not impact searches for massive binary black holes but could influence neutron star analyses. I wonder whether future subsolar searches with experiments like the Einstein Telescope and Cosmic Explorer might also be affected by these effects.

- In Sec. 2.2, they comment that glitches exhibit characteristics similar to CBC signals with extreme mass ratios and large spins. Do they have a physical understanding of such similarity?

- In Sec. 2.3, the authors say that detection and parameter estimation of overlapping signals are not a significant concern, without however providing a motivation. They add few comments on line 252 and 253, but it would be useful to expand the above paragraph. Similarly, on line 262 the authors state that when results are combined on a population level, the biases tend to smoothen out. This sentence seem to be in contrast with the last three sentences (line 275-278), where they comment that population analyses may accumulate biases (see also first line on page 23). A clarification on this point is needed.

- In Sec. 2.5, they comment that parameters that suffer large biases are those related to the GW amplitude. The paper would benefit from further explanation of this property.

- In Sec. 3.1.1, the authors comment about the impact of standard astrophysical scenarios on eccentricity. The authors could mention as well the role of accretion and dynamical friction on eccentricity, see https://arxiv.org/pdf/2010.15151.

- In Sec. 3.1.2, they state that not including subdominant tidal effects, such as dynamical tides, can lead to biases in the estimation of tidal parameters. I was wondering if something similar could happen for nonlinear tides (mentioned in their Ref. [184]). Adding further details could make the paragraph more comprehensive. In the same paragraph, they also mention the role of universality relations (which hold to a certain accuracy) in GW data. Does this intrinsic (subpercent-) accuracy impact on the analyses?

- In Sec. 3.1.4, the authors talk about testing of no-hair theorems. It would be useful if they could stress the distinction between primary and secondary hair, to be more precise.

- In Sec. 3.2.1, they discuss about memory effects, see end of page 15, Table 1 (cross on O4) and Sec. 5.2, on a single event basis. The paper would benefit of further additions about similar considerations on a population-based analysis of memory effects and the so-called "stacking" procedure. Would similar conclusions hold on a population level?

- On lines 626-630, the authors propose that a probability distribution modelled through a multidimensional Gaussian mixture could account for uncertainty estimates due to sub-optimal fits, and could mitigate waveform systematics at high SNR. I was wondering if such approach could in principle loose any putative non-Gaussian information in the data. Moreover, the sentence on line 666 is a bit unclear, may they rewrite it?

- In Sec. 4.2, the authors discuss environmental effects around binary black holes, not considering clouds of ultralight bosons as environment, which are instead thoroughly discussed in Sec. 4.3.2 from line 914. I was wondering if the paper would benefit of moving that discussion to the above section, where environments are discussed. However, if the authors prefer to keep them there, I would at least suggest adding a sentence about them and a reference to the paragraph in Sec. 4.3.2 (even though I do not see why these clouds should be better described as mimickers and not as environments).

- In Sec. 4.3.2, can they clarify why black hole mimickers with no horizon generally have a much smaller GW absorption cross section than black holes? Furthermore, they could also mention tidal disruption as a changing feature.

- In the paragraph starting on line 951, I was wondering if mass gap events could fall within the category of events detected in regions of the parameter space disfavored by astrophysical population models (even though hierarchical mergers could falsify this possibility).

- On line 1092, the authors forgot to add $g/cm^3$ after $\rho > 10$.

- One possible suggestion for the authors is to highlight key sentences for each of the potential false violations, helping readers to quickly grasp the main takeaways.

---

## Round 2 · Referee Report · Anonymous (Referee 1) · 2024-12-9

Report

Dear Editor,

I thank the authors for carefully addressing my comments. I am satisfied with their additions to the manuscript and I am glad to recommend the paper for publication in SciPost.

Recommendation

Publish (easily meets expectations and criteria for this Journal; among top 50%)

---

## Round 2 · Author Response

Dear Editor,

We thank the referee for carefully going through our paper and providing detailed and valuable suggestions to improve its presentation. We have addressed each of the referee’s comments and made changes (in blue text) to the draft wherever appropriate. Please find our response to the referee below.

Best,
Authors

---

## Round 2 · List of Changes

• In Sec. 2.1, it would be helpful if the authors could include examples of both instrumental and environmental sources of non-stationarity. Additionally, they mention that these effects should not impact searches for massive binary black holes but could influence neutron star analyses. I wonder whether future subsolar searches with experiments like the Einstein Telescope and Cosmic Explorer might also be affected by these effects.

We have added examples of sources of non-stationary noise. Though there is no concrete study on the effect of non-stationary on subsolar mass binary searches, one would expect that they will also be affected due to their long duration in the sensitivity band of next-generation GW detectors. We have added a sentence in this regard.

• In Sec. 2.2, they comment that glitches exhibit characteristics similar to CBC signals with extreme mass ratios and large spins. Do they have a physical understanding of such similarity?

We have added an explanation of this similarity (as provided in Ashton et al.) after that sentence. Since mass ratios considered in Ashton et al. (10.1088/1361-6382/ac8094) are not unequal enough to be really counted as "extreme" (though the authors do use this word), we have replaced “extreme” with “large” in Sec. 2.2 and in "Precession and Higher-order Modes" paragraph of Sec. 5.2.

• In Sec. 2.3, the authors say that detection and parameter estimation of overlapping signals are not a significant concern, without however providing a motivation. They add few comments on line 252 and 253, but it would be useful to expand the above paragraph. Similarly, on line 262 the authors state that when results are combined on a population level, the biases tend to smoothen out. This sentence seem to be in contrast with the last three sentences (line 275-278), where they comment that population analyses may accumulate biases (see also first line on page 23). A clarification on this point is needed.

We have expanded the discussion to clarify that overlapping signals are not an issue for searches and parameter estimation. Thank you for pointing out the contradictory statements about combining events; we have corrected this. Additionally, we have restructured the text in the first three paragraphs of Sec. 2.3 to improve readability and flow.

• In Sec. 2.5, they comment that parameters that suffer large biases are those related to the GW amplitude. The paper would benefit from further explanation of this property.

We have expanded the discussions in Sec. 2.5.

• In Sec. 3.1.1, the authors comment about the impact of standard astrophysical scenarios on eccentricity. The authors could mention as well the role of accretion and dynamical friction on eccentricity, see https://arxiv.org/pdf/2010.15151.

We have added the text about the role of accretion and dynamical friction on eccentricity, citing the above reference.

• In Sec. 3.1.2, they state that not including subdominant tidal effects, such as dynamical tides, can lead to biases in the estimation of tidal parameters. I was wondering if something similar could happen for nonlinear tides (mentioned in their Ref. [184]). Adding further details could make the paragraph more comprehensive. In the same paragraph, they also mention the role of universality relations (which hold to a certain accuracy) in GW data. Does this intrinsic (subpercent-) accuracy impact on the analyses?

Since the nonlinear tides can change the GW phase by 10-20% at 1000 Hz even at Newtonian order, we would expect they could lead to biases in the tidal (and other source) parameters. NR simulations include nonlinear tidal effects and there are studies (e.g., doi:10.1103/PhysRevD.98.08406) which show that waveform models that do not agree well enough with NR waveforms lead to biases. However, as far as we know there has not been any study that quantifies the effect of nonlinear tides solely. Since we mention nonlinear tides along with other long list of tidal effects, we don’t think it will be appropriate to expand the discussion just on nonlinear tides, not without discussing all other effects also in detail. However, we have moved the sentence “Additionally, effects of spins on dynamical tides .... further investigations” to the third paragraph where we discuss dynamical tides and feel that it will be more suited over there. We have also added a sentence to comment on the subpercent accuracy of universality relations on data analysis.

• In Sec. 3.1.4, the authors talk about testing of no-hair theorems. It would be useful if they could stress the distinction between primary and secondary hair, to be more precise.

Thanks for the suggestion. We have added a couple of sentences in Sec. 3.1.4 to clarify the distinction between primary and secondary hair.

• In Sec. 3.2.1, they discuss about memory effects, see end of page 15, Table 1 (cross on O4) and Sec. 5.2, on a single event basis. The paper would benefit of further additions about similar considerations on a population-based analysis of memory effects and the so-called “stacking” procedure. Would similar conclusions hold on a population level?

We have added a sentence about detecting memory in individual sources and a population using the stacking method in Sec. 3.2.1. Though we think that parameter estimation analyses at the population level in future detectors would be biased by neglect of the memory, as far as we can tell, there is no study that shows this explicitly. We have added one more sentence in this regard.

• On lines 626-630, the authors propose that a probability distribution modelled through a multidimensional Gaussian mixture could account for uncertainty estimates due to sub-optimal fits, and could mitigate waveform systematics at high SNR. I was wondering if such approach could in principle loose any putative non-Gaussian information in the data. Moreover, the sentence on line 666 is a bit unclear, may they rewrite it?

The proposed method employs Gaussian mixtures to model the probability distribution for higher-order (unknown) PN coefficients that are calibrated to NR waveforms, rather than directly modeling the observational data itself. These coefficients primarily impact the late inspiral and merger portions of the signal, which are not known analytically and therefore require calibration to NR simulations. The earlier inspiral and ringdown stages of the signal remain unaffected by this approach. While it is possible that this method could obscure small deviations from GR occurring around the merger phase, the width of these waveform modeling uncertainties can be estimated in advance across the parameter space. As a result, significant GR deviations in the merger phase (large enough relative to the estimated modeling uncertainties) would likely still be detectable. We have added a sentence about it. We have also rewritten the sentence around line 666 (now L699-L703) to improve clarity.

• In Sec. 4.2, the authors discuss environmental effects around binary black holes, not considering clouds of ultralight bosons as environment, which are instead thoroughly discussed in Sec. 4.3.2 from line 914. I was wondering if the paper would benefit of moving that discussion to the above section, where environments are discussed. However, if the authors prefer to keep them there, I would at least suggest adding a sentence about them and a reference to the paragraph in Sec. 4.3.2 (even though I do not see why these clouds should be better described as mimickers and not as environments).

Since boson clouds depend on the binary’s properties (and the universal properties of the boson) and are not something external to the binary, we treated them differently from other environmental effects. Additionally, the deviations from the GR waveform due to boson clouds are similar to the ones due to black hole mimickers. That’s why we kept the boson clouds discussion in Sec. 4.3.2. But now following the referee’s suggestion, we have moved the boson clouds text from Sec. 4.3.2 to Sec. 4.2 (with some modifications) and highlighted the different natures of boson clouds and standard environmental effects.

• In Sec. 4.3.2, can they clarify why black hole mimickers with no horizon generally have a much smaller GW absorption cross section than black holes? Furthermore, they could also mention tidal disruption as a changing feature.

Since recent work has shown that it is possible that neutron stars can have larger GW absorption than black holes if they have a sufficiently large shear viscosity, we have now rewritten the discussion to reflect this, and this discussion also clarifies why black hole mimickers such as standard models of boson stars do not have significant GW absorption. We also now mention tidal disruption as a possible difference between binaries of black hole mimickers and binaries of black holes.

• In the paragraph starting on line 951, I was wondering if mass gap events could fall within the category of events detected in regions of the parameter space disfavored by astrophysical population models (even though hierarchical mergers could falsify this possibility).

This is definitely a possibility for the near-future in GW astronomy, and would likely be an indication that improvements to the underlying astrophysical population model are required. While current models for the astrophysical population do not introduce more complex physics such as hierarchical merger subpopulations, as more observations are made, more complex astrophysical models will be required to accurately test GR. To this end, we’ve added a sentence to the paragraph providing hierarchical mergers as an example.

• On line 1092, the authors forgot to add g/cm3 after ρ > 10.

Done.

• One possible suggestion for the authors is to highlight key sentences for each of the potential false violations, helping readers to quickly grasp the main takeaways.

It might be difficult to do this given that there are so many effects and we do not discuss each of them in equal detail (since we do not have much literature on some of the causes). The point of Sec. 5 (and Table 1) is precisely to provide readers with a comprehensive summary of all effects discussed in the paper. We, therefore, prefer not to highlight any sentences in the paper.

---

## Editorial Decision

published